# Towards Decision Focused Learning for Sparse and Weakly Supervised Environments

## Abstract

Decision-focused learning (DFL) integrates machine learning and optimisation by training predictive models to directly optimise decision quality. However, DFL typically depends on access to accurate ground-truth targets such as objective function parameters or optimal decisions: This is often infeasible in real-world settings where only sparse, binary feedback on decision outcomes is available.

This work takes first steps towards formalising Decision-Focused Learning for settings where observed data is limited to such binary feedback. We propose a preliminary DFL framework that learns latent user preferences from weakly supervised binary feedback on decision outcomes. The novelty of our approach lies is in a) a ground-truth-free, differentiable surrogate loss that maps binary evaluations to decision outcomes, and b) a novel meta-learning mechanism that learns latent user preference patterns and transfers this knowledge between users to mitigate challenges due to per-user data sparsity.

Our experiments suggest that this framework can reduce decision regret by 20-fold and achieves convergence with $2.4 - 4\times$ fewer data points than standard predict-then-optimise baselines. On a novel hyper-sparse real-world trip-planning feedback dataset, we show the model's ability to extract user-preference clusters from sparse data ($\approx 1$ interaction/user). We also evaluate our model in cold-start recommendation settings and show that our decision loss correctly prioritises ranking quality, achieving 9.5% higher nDCG@5 than the baseline despite 14.6% higher MSE. The aim of this work is to broaden the applicability of DFL and explore its potential in weakly supervised data-sparse regimes, with future work extending non-linear user-preference structure.

## 1 Introduction

Decision-Focused Learning (DFL) (1) has gained recent interest for solving decision-making problems under uncertainty, a challenge prevalent in real-world applications such as transportation, finance, and recommender systems. Unlike Prediction-Focused Learning (PFL) paradigm—a two-stage process that first predicts uncertain parameters and then solves an optimisation problem to derive decisions—DFL integrates prediction and optimisation into a unified framework (2).

A key feature of DFL methods is that they typically estimate the parameters of the underlying optimisation problem by assuming access to a correlated dataset containing the optimisation problem parameters as target variables (3–7). These maybe presented either as the objective function's true parameters (3–6) or true optimal decisions derived from unknown parameters (7). However, there are many real-world scenarios —e.g. transportation, health care or personalised recommendation systems —where such supervision is unavailable, invasive, or costly to obtain.

This work extends DFL to a weak-supervision setting in multi-agent scenarios where we only observe sparse binary feedback on predicted decisions (e.g., a like/dislike of a route suggestion, a recommended movie, or an LLM output). We demonstrate that when such binary evaluations of decision outcomes are available in place of full ground-truth supervision, the underlying DFL problem can still be addressed: rather than learning from exact parameters or optimal solutions, the learner can be trained to predict decisions that are likely to be deemed *acceptable* (within a specified tolerance of optimality). This reduction enables DFL to be applied to domains where only binary evaluations are observable.

Relying on binary evaluations, however, introduces two main challenges. First, the signal is inherently coarse: a positive label indicates that a decision is acceptable within tolerance, but does not imply optimality, while a negative label provides little guidance on how far the decision is from the true optimum. Second, such feedback is typically sparse in the real-world, leading to cold-start issues. While cold-start problems are well-studied in recommender systems (8), our setting is more demanding: each "recommendation" here corresponds not to a static item but to the solution of a parameterised optimisation problem, making standard approaches inapplicable.

To address these challenges, we propose a framework that unifies **decision-focused learning (DFL)** with **meta-learning** via a bi-level optimisation structure. This framework bridges parameter estimation in sparse data regimes with the goal of optimising decision quality under weak supervision. Our framework is characterised by two key features:

1. **Decision loss for weak supervision**: To overcome the coarse nature of a binary feedback, we design an approximate differentiable decision loss. We achieve this by first defining an "acceptability predicate" (Section 3.1) to formally model the feedback as a function of the latent ground truth parameter. We then design a differential surrogate to the true decision loss as a function of the binary feedback. This allows our model to be trained end-to-end with the combinatorial solver to optimise for decision quality.

2. **Meta-learning for cold-start adaptation**: Based on the assumption that users exhibit latent preference groupings (9)—for instance, a group of travellers prioritising affordability or commuters valuing speed— our framework identifies and leverages latent preference structures across users (10) to make robust decisions even when individual data is scarce. This approach has broad implications, from adapting transportation systems to evolving commuter priorities to aligning AI agents with user preferences under minimal supervision.

Unlike standard prediction pipelines that minimise a prediction-error loss (e.g., cross-entropy for labels or RMSE for ratings), our DFL framework trains a prediction model alongside a combinatorial solver via a differentiable surrogate decision loss (see Section 4): the predictor outputs parameters for the solver, the solver (or a differentiable approximation of it) produces a decision, and the surrogate loss measures decision quality. Therefore, we do not train the model to predict a label or score, but to produce parameters that results in high-quality, task-specific decisions (for example, a custom route or allocation) from the downstream optimisation problem. This coupling of prediction and optimisation differentiates DFL from standard classification or recommender pipelines. Importantly, the solution space here is dynamic: each "recommended item" is not drawn from a fixed, finite catalogue, but is instead the solution to a parametrised optimisation problem over a potentially infinite set of feasible solutions.

To illustrate the application of our framework, we introduce personalised public transit trip planning as a weakly supervised DFL problem. User preferences (e.g., prioritising cost versus speed) are inferred from limited feedback on route recommendations (see an illustrative example in Appendix B) We validate our findings by using real-world feedback data (Appendix D.1.2), marking this work as the first real-world application of DFL in personalised trip planning. Additionally, we also evaluate our framework in cold-start recommendation. Though not primarily our objective, our experiments in hyper-sparse regimes (fewer than five data points per user) show that our framework performs competitively against established architectures (Section 6) using a simple linear model as the predictor in the DFL framework. Note that the linear predictor is trained to minimise decision loss; in contrast to vanilla logistic regression which minimises prediction loss on observed feedback.

1. **Formalising DFL under Binary Supervision**: We propose a general framework for Decision-Focused Learning in settings where ground-truth parameters are unavailable and supervision is limited to binary feedback on decision outcomes. We formulate a decision loss function using only this feedback which serves as a bounded approximation of the true decision regret (Appendix C). We then demonstrate the applicability of this framework through a viable implementation: a novel differentiable surrogate loss within a bi-level optimisation framework.

2. **Novel Meta-Learning Approach in Cold-Start Setting**: Learns latent user preference patterns (e.g., travellers prioritising affordability), and transfers knowledge within clusters to reduce per-user data requirements by up to 75% in empirical tests.

3. **Real-world validation and dataset**: Evaluates the personalised trip planning model on a novel hyper-sparse feedback dataset of 20k interactions collected through a trip planner from over 12k

unique users. This dataset exemplifies how lightweight feedback mechanisms empower smaller organisations to adopt DFL.

## 2 RELATED WORKS

Decision-Focused Learning (DFL) trains predictive models to directly optimise downstream decision objectives rather than standard two-stage prediction-and-solve pipelines (11; 2; 4). Most existing methods assume full supervision via known cost parameters or optimal decision labels (12; 7), but many real-world applications only provide sparse binary feedback on solution quality (1; 3). Related fields—such as reinforcement learning and bandit optimisation—address learning under limited feedback (13–15), yet typically do not evaluate decision quality on downstream combinatorial tasks. In this work, we model a DFL problem in a multi-agent scenario where each agent only provides sparse binary feedback on the decision outcomes as ground truth (9; 16–18); bridging decision-focused learning, combinatorial optimisation, and weak supervision. For a more comprehensive survey of related works, see Appendix A.

## 3 WEAKLY SUPERVISED DFL: MODEL DEFINITION

Consider a decision-making task where the objective is to solve an optimisation problem that maximises a utility function $u(x, \theta)$ given a set $\mathcal{X}$ of feasible solutions. The parameters $\theta \in \mathbb{R}^d$ are not known a priori. The optimal choice $x^*(\theta) \in \mathcal{X}$ is defined as:

$$x^*(\theta) = \arg \max_{x \in \mathcal{X}} \ u(x, \theta), \tag{1}$$

The goal of a DFL problem is to learn a parameter estimate $\hat{\theta}$ that results in optimal decisions through end-to-end optimisation. In the DFL setup (1), this done by minimising *regret* which quantifies the utility loss from decisions made with estimated parameter $\hat{\theta}$ (i.e. $x^*(\hat{\theta})$) versus the true optimum $x^*(\theta)$, when evaluated using the true parameter $\theta$:

$$\text{Regret}(\hat{\theta}, \theta) = u(x^*(\theta), \theta) - u(x^*(\hat{\theta}), \theta) \tag{2}$$

However, minimising regret through gradient-based methods is non-trivial as a) $\mathcal{X}$ could be discrete, and b) minimising the regret function involves differentiating through the $\arg \max$ operator via $x^*(\hat{\theta})$. To avoid the consequent stability problems, researchers generally minimise a surrogate loss function that either is a relaxation of the regret function (4; 19; 20) or an approximation of it in the same decision space (6; 2; 21). For instance, Mulamba et al. (3) define noise contrastive estimation (NCE) surrogate loss where they consider a subset of decision variables as negative examples. Shah et al. (6) replace surrogate loss altogether by learning task-specific loss functions.

### 3.1 WEAK SUPERVISION PARADIGM: BINARY LABELS ON DECISION OUTCOMES

DFL models are generally trained using labelled datasets $\mathcal{D} = \{(z_i, \theta_i)\}_{i=1}^N$, where features $z_i$ are mapped to ground-truth parameters $\theta_i$. In this work, we instead explore a scenario where $\theta_i$ are either not available or are unobserved. We consider a weakly supervised setting where our dataset $\mathcal{D} = \{(z_i, x_i^*, f_i)\}_{i=1}^N$, consists of triples where $z_i$ is a mapping to the discrete decision space $\mathcal{X}_{z_i}$ and $x_i^* = x^*(\theta')$ is the solution generated by an initial parameter estimate $\theta'^1$. The binary feedback $f_i = f(x_i^*, \theta_i) \in \{0, 1\}$ is obtained by evaluating $x_i^*$ under the latent true parameter $\theta_i$. Formally, feedback is defined as an *acceptability predicate* for a given decision $x$:

$$f(x, \theta) = \mathbf{1}\{u(x, \theta) \geq \tau \cdot u(x^*(\theta), \theta)\} \tag{3}$$

where $x^*(\theta)$ is the true optimal decision. The tolerance factor $\tau \in (0, 1]$ quantifies the user's threshold for accepting a sub-optimal solution.

We consider $\tau$ as a "satisficing" metric (22): a positive feedback $f = 1$ signals only near-optimality (a "good-enough" solution) according to a user's threshold, and not true optimality. We also note that $\tau$ is not a trainable hyperparameter; Equation 3 is introduced solely to interpret the meaning of

---

[1]The estimate $\theta'$ is the initial parameter chosen for all users prior to any data collection

the binary feedback. In our setting, $\tau$ is treated as an intrinsic characteristic of the dataset, and we only observe $f(x, \theta) \in \{0, 1\}$.

Recall that the objective of a DFL problem is to estimate the parameter $\hat{\theta}$ that minimises regret (equation 2), typically via a surrogate loss function. However, the absence of the true parameter ($\theta$) prevents the regret (or its surrogate) from being evaluated, rendering the objective intractable. We, therefore, propose a novel surrogate objective as a function of the observable signal $f_i$: to estimate a parameter $\hat{\theta}$ that maximises the probability of the predicted outcome ($x^*(\hat{\theta})$) being 'acceptable', i.e. $P[f(x^*(\hat{\theta}), \theta) = 1]$.

We formally prove in Appendix C that maximising this probability tightens the upper bound on the expected regret. This establishes a general, problem-independent reduction for DFL under binary supervision: the intractable goal of minimising regret can be replaced by the tractable surrogate goal of maximising the probability of an acceptable outcome. The remainder of this section, and Section 4, focuses on the decision-making framework for an individual agent, which we later generalise to the multi-agent setting in Section 5.

## 3.2 OPTIMISATION PROBLEM

Based on the discussion in Section 3.1, we define utility $u(x, \theta)$ as probability of obtaining a positive feedback for some input $x \in \mathcal{X}$ and latent parameter $\theta$. Consequently, the DFL optimisation problem in equation 1 is redefined as finding a solution $x^*(\theta)$ for given $\theta$, that maximises the probability of receiving positive feedback $f(x^*, \theta)$, i.e.:

$$x^*(\theta) \;=\; \arg\max_{x \in \mathcal{X}_z} u(x, \theta) \;=\; \arg\max_{x \in \mathcal{X}_z} P(f(x, \theta) = 1 \mid x, \theta) \qquad (4)$$

The next challenge is to model the distribution of binary feedback. Since the feedback on a decision is inherently binary, we require a probabilistic mapping from latent scores to observed outcomes. We adopt a sigmoidal function with a noisy linear input, which provides both analytical simplicity and implementation stability. More complex non-linear mappings are possible, but under sparse binary feedback they are more prone to overfitting. In contrast, the linear-plus-sigmoid formulation offers a robust baseline that remains empirically effective in our experiments. Hence,

$$u(x, \theta) = P(f(x, \theta) = 1 \mid x, \theta) = \sigma\left(\frac{\theta^\top x + b}{s}\right), \qquad (5)$$

where $\sigma(z) = \frac{1}{1+e^{-z}}$ is the sigmoid function, and $s$ and $b$ are tuned scale and bias hyper-parameters respectively.

## 4 PARAMETER ESTIMATION

Let $\hat{\theta}$ be the estimate of the latent parameter $\theta$. If $x^*(\hat{\theta})$ is the predicted optimal solution through equation 4, we redefine the *regret* in equation 2 as:

$$\text{Regret}(\theta, \hat{\theta}) = u(x^*(\theta), \theta) - u(x^*(\hat{\theta}), \theta) = \sigma\left(\frac{\theta^\top x^*(\theta) + b}{s}\right) - \sigma\left(\frac{\theta^\top x^*(\hat{\theta}) + b}{s}\right) \qquad (6)$$

However, we cannot evaluate $\text{Regret}$ as it requires a) $x^*(\theta)$ which is unobserved, and b) computing $\theta^\top x^*(\hat{\theta})$, which is not possible as $\theta$ is also unobserved. We thus design a surrogate loss independent of the parameter $\theta$ that a) predicts an accurate feedback $f(x^*(\hat{\theta}), \hat{\theta}) \approx f(x^*(\hat{\theta}), \theta)$ for the solution $x^*(\hat{\theta}) \in \mathcal{X}$, and b) that $\hat{\theta}$ ensures correct ordering of the utilities $u(x, \hat{\theta}) \, \forall \, x \in \mathcal{X}$, i.e.

$$u(x_\alpha, \theta) > u(x_\beta, \theta) \implies u(x_\alpha, \hat{\theta}) > u(x_\beta, \hat{\theta}) \; \forall x_\alpha, x_\beta \in \mathcal{X}$$

To address these dual objectives, we consider two key observations. First, since we model the utility $u(x, \theta)$ as a sigmoid function in equation 5, the requirement for accurate feedback prediction naturally aligns with the Binary Cross Entropy (BCE) Loss:

$$\mathcal{L}_{\text{BCE}} = -\sum_{i=1}^{|\mathcal{X}_z|} \Big[ f_i \log(u(x_i, \hat{\theta})) + (1 - f_i) \log(1 - u(x_i, \hat{\theta})) \Big]. \qquad (7)$$

Secondly, the logistic noise assumption motivates the use of a ranking-oriented loss, particularly Pairwise Logistic Loss (PLL). Originally described in RankNet (23), PLL penalises mis-ordered pairs by applying a logistic penalty to the difference in their predicted scores, thereby encouraging higher-scored items to align with higher true preferences. Formally, PLL is defined as:

$$\mathcal{L}_{\text{PLL}} = \sum_{i=1}^{|\mathcal{X}_z|} \sum_{j=1}^{|\mathcal{X}_z|} \mathbb{I}[f^i > f^j] \log\left(1 + \exp\left(-(u(x_i, \hat{\theta}) - u(x_j, \hat{\theta}))\right)\right), \tag{8}$$

where $(u(x_i, \hat{\theta}), u(x_j, \hat{\theta}))$ are logistic scores, $f^i = f(x_i, \theta)$ and $f^j = f(x_j, \theta)$ are the corresponding targets, and $\mathbb{I}[.]$ is an indicator function.

The direct application of PLL and BCE to our setting faces a critical limitation: feedback $f(x^*(\theta'), \theta)$ is observed only for the chosen solution $x^* = x^*(\theta')$, with no labels available for other candidates $x \in \mathcal{X}_z \setminus \{x^*\}$. This sparsity of feedback prevents a) pairwise comparisons $\mathbb{I}\left[f^j > f^k\right]$ in equation 8, and b) calculation of the BCE loss $\forall\ x_j \in \mathcal{X}_z \setminus \{x^*\}$. To reconcile with these constraints, we reformulate the two loss functions. Our proposed surrogate loss incorporates the optimisation structure of $x^*(\hat{\theta})$ while generalising feedback inferences to unobserved candidates.

### 4.1 SURROGATE LOSS

Our approach to the aforementioned challenges is guided by two motivating simplifications:

1. **Optimal solution dominance:** Positive feedback for a solution $x^*$ implies it dominates all other options in $\mathcal{X}_z$, while negative feedback implies sub-optimality:
   - $f(x^*, \theta) = 1 \implies u(x^*, \hat{\theta}) \geq u(x_j, \hat{\theta}) \quad \forall x_j \in \mathcal{X}_z \setminus \{x^*\}$
   - $f(x^*, \theta) = 0 \implies u(x^*, \hat{\theta}) \leq u(x_j, \hat{\theta}) \quad \forall x_j \in \mathcal{X}_z \setminus \{x^*\}$
2. **Pseudo-labelling for BCE:** For decisions with unobserved feedback, we adopt conservative pseudo-labels consistent with our binary formulation equation 3:
   - If $f(x^*, \theta) = 1$, we treat $x^*$ as the sole positive example and all others as negative.
   - If $f(x^*, \theta) = 0$, we label all solutions in $\mathcal{X}_z$ as negative: no satisfactory choice exists.

While these simplifications are strict, we emphasise that they are motivated by the structure of our framework described in Section 3.1, and in turn, motivate the design of our surrogate loss function described in Section 4. Our experiments in Section 6 demonstrate that these approximations, while possibly imperfect, do not degrade performance in practice suggesting that they capture sufficient structure to guide parameter estimation effectively, even under weak supervision and data sparsity. Consequently, we reformulate the PLL and BCE loss function.

**Modified Pairwise Logistic Loss:** For each instance $(z_i, x_i^*, f_i) \in \mathcal{D}$, we define the *modified pairwise logistic loss* (MPLL) as:

$$\mathcal{L}_{\text{MPLL}}(\hat{\theta}) = \sum_{x_j \in \mathcal{X}_{z_i} \setminus \{x_i^*\}} \log\left(1 + \exp\left(-\gamma_{ij}\right)\right), \tag{9}$$

where $\quad \gamma_{ij} = (2\delta_i - 1)\left(u(x_i^*, \hat{\theta}) - u(x_j, \hat{\theta})\right)$ and $\quad \delta_i = f(x_i^*, \theta_i)$

The term $(2\delta_i - 1)$ ensures correct ranking direction: enforcing $u(x_i^*, \hat{\theta}) \geq u(x_j, \hat{\theta}))$ for positive feedback $((2\delta_i - 1) = +1)$ and $u(x_i^*, \hat{\theta}) \leq u(x_j, \hat{\theta})$ for negative feedback $((2\delta_i - 1) = -1)$.

**Modified BCE Loss:** We first extend the BCE loss by assigning pseudo-feedback for all candidates $(x_j \in \mathcal{X}_{z_i})$ as:

$$y_j = \delta_i \mathbb{I}[x_j = x_i^*] = \begin{cases} 1, & x_j = x_i^* \text{ and } \delta_i = 1, \\ 0, & \text{otherwise.} \end{cases}$$

The BCE loss with these pseudo-feedbacks is then:

$$\mathcal{L}_{\text{BCE}_*}\left(\hat{\theta}\right) = -\sum_{x_j \in \mathcal{X}_{z_i}} \left[\delta_i \mathbb{I}[x_j = x_i^*] \log u(x_j, \hat{\theta}) + (1 - \delta_i \mathbb{I}[x_j = x_i^*]) \log(1 - u(x_j, \hat{\theta}))\right] \tag{10}$$

### 4.1.1 Surrogate Loss Definition: Balancing Ranking and Prediction Accuracy

Defining $\mathcal{L}_{\text{MPLL}}$ and $\mathcal{L}_{\text{BCE*}}$, our framework must simultaneously ensure accurate preference rankings under weak supervision and feedback predictions under data sparsity. Therefore, we define surrogate loss function as the sum:

$$\mathcal{L}_{\text{surrogate}} = \mathcal{L}_{\text{MPLL}} + \lambda\,\mathcal{L}_{\text{BCE*}}, \qquad (11)$$

where the regularisation parameter $\lambda$ balances the relative importance of ranking consistency ($\mathcal{L}_{\text{MPLL}}$) versus feedback prediction ($\mathcal{L}_{\text{BCE*}}$).

As claimed initially, the surrogate loss in (11) is free of the ground truth parameter $\theta$. We only require $\delta_j$ which is the observable ground-truth feedback. Importantly, our surrogate loss is defined over the 'acceptability' of solver-generated decisions. The training produces parameters $\hat{\theta}$ that steer the optimiser toward making high-quality decisions which are more likely to receive positive feedback. Unlike conventional recommender or classification systems, which learn from explicit item labels or user–item rating matrices over a fixed catalogue, the DFL framework learns parameter mappings that directly improve the quality of the decisions produced by an optimisation solver. Furthermore, calculating the gradient of equation 11 does not require differentiating through an $\arg\max$ operator as $\delta_j = f\left(x_i^*, \theta\right)$ is a known parameter and all other parameters are independent of $\theta$.

## 5 Collaborative Training

Our previous formulation assumes a single decision-maker with fixed preferences. We now consider $N$ interacting agents, each with their latent utility parameters $\theta^i \in \mathbb{R}^d$. Let $\mathcal{U} = 1, 2, \ldots, N$ define the embedding matrix.

$$\hat{\Theta} = [\hat{\theta}^1, \hat{\theta}^2, \ldots, \hat{\theta}^N] \ \in \ \mathbb{R}^{N \times d},$$

where $\hat{\theta}^i$ denotes estimated preference vector of agent $i$. We further assume a low-rank structure: $\exists$ a small basis set $\mathcal{B}$ ($|\mathcal{B}| \ll |\mathcal{U}|$) such that each $\hat{\theta}^i$ is a noisy perturbation of some $\mathcal{B}_i \in \mathcal{B}$. For example, commuters often share constraints on fare, travel time, or mode preferences in trip planning (17). This induces (a) knowledge transfer via shared updates among similarly-parametrised agents, and (b) cluster training to jointly optimise agents with aligned preferences, improving data efficiency.

**Agent Neighbourhood** To capture "similarity" among agents, we define each agent's neighbourhood:

$$\mathcal{N}_i = \{\, j \in \mathcal{U} \ | \ \epsilon_0 \le d(\hat{\theta}^i, \hat{\theta}^j) \le \epsilon \,\},$$

where $d(\cdot, \cdot)$ is a distance metric (e.g. Euclidean), $\epsilon_0 > 0$ excludes trivial self-matches, and $\epsilon$ sets the maximum radius. The choice of distance metric $d$ is flexible, provided it reflects behavioural alignment between agents (e.g. the difference in outputs). Feedback provided by agent $i$ contributes to the training of all agents $j \in \mathcal{N}_i$, with influence inversely weighted by $d(\hat{\theta}^i, \hat{\theta}^j)$. We call this scheme *collaborative training*. Computing $\mathcal{N}_i$ can be a computational bottleneck with non-linear models or high-dimensional vector spaces. We discuss this issue in Appendix E.1.1.

### 5.1 Collaborative Optimisation Problem

For the population $\mathcal{U}$, we define the composite dataset $\mathcal{D}_{\mathcal{U}} = \bigcup_{i \in \mathcal{U}} \mathcal{D}_i$, where $\mathcal{D}_i$ contains feedback instances from agent $i$. Each instance comprises the agent identifier, query parameters, chosen solution, and binary feedback:

$$\mathcal{D}_{\mathcal{U}} = \left\{ \left(i, z_{k_i}, x_{k_i}^*, f(x_{k_i}^*, \theta^i)\right) \ \middle| \ i \in \mathcal{U}, \ k_i \in \{1, \ldots, |\mathcal{D}_i|\} \right\}. \qquad (12)$$

The collaborative framework generalises the utility function in equation 5 by integrating neighbourhood consensus: the weighted average of the neighbourhood predictions. For agent $i \in \mathcal{U}$, the latent utility score $\psi_i(x)$ is:

$$\psi_i(x) = (1-\alpha)\hat{\theta}^{i\top} x + \underbrace{\alpha \cdot \frac{\sum_{j \in \mathcal{N}_i} \frac{\hat{\theta}^{j\top} x}{d(\hat{\theta}^i, \hat{\theta}^j)}}{\sum_{j \in \mathcal{N}_i} \frac{1}{d(\hat{\theta}^i, \hat{\theta}^j)}}}_{\text{Neighbourhood-weighted consensus}} \qquad (13)$$

Figure 1: Illustration of DFL + Collaborative Training Architecture for a personalised trip planning problem. For a query $z$ by a user $i$: a) the optimal decision $x^*$ is obtained through all the predicted weight vectors in neighbourhood $\mathcal{N}_i$, and b) the gradient update is done $\forall j \in \mathcal{N}_i$

where $\alpha \in [0, 1]$. The resultant optimisation problem for an agent $i \in \mathcal{U}$ and query $z_{k_i}$ then becomes:

$$x^*(\theta^i) = \arg \max_{x \in \mathcal{X}_{z_{k_i}}} \sigma \left( \frac{\psi_i(x) + b}{s} \right),$$
(14)

Notably, the collaborative utility in equation 13 closely resembles an attention mechanism. In this analogy, the agent's preference vector $\hat{\theta}^i$ serves as the "query" while the neighbours' preference vectors $\hat{\theta}^j$ act as "keys". The resulting utilities, $\hat{\theta}^{j\top} x$ are the "values" and the weight $\frac{\frac{1}{d(\hat{\theta}^i, \theta^j)}}{\sum_{j \in \mathcal{N}_i} \frac{1}{d(\hat{\theta}^i, \hat{\theta}^j)}}$ is the attention score, quantifying the relevance of each neighbour. However, a formal replacement of the distance-based neighbourhood with a learned attention mechanism is not explored in this paper and remains an interesting avenue for future work.

## 5.2 COLLABORATIVE SURROGATE LOSS

Under the collaborative training framework, we reformulate MPLL and BCE loss functions by replacing individual utilities (Equation 5) with neighbourhood-informed collaborative utilities:

$$u_{collab}(x, \theta) = \sigma \left( \frac{\psi(x) + b}{s} \right)$$

**Collaborative Pairwise Loss**

$$\mathcal{L}_{\text{CMPLL}}(\hat{\Theta}) = \sum_{x_j \in \mathcal{X}_{z_i} \setminus \{x_{k_i}^*\}} \log \left( 1 + \exp \left( -\tilde{\gamma}_{ij} \right) \right),$$
(15)

where $\quad \tilde{\gamma}_{ij} = (2y_{k_i} - 1) \left[ \sigma \left( \frac{\psi_i(x_{k_i}^*) + b}{s} \right) - \sigma \left( \frac{\psi_i(x_j) + b}{s} \right) \right]$ and $\quad y_{k_i} = f(x_{k_i}^*, \theta_i)$

**Collaborative BCE Loss**

$$\mathcal{L}_{\text{CBCE*}} = - \sum_{x_j \in \mathcal{X}_{z_{k_1}}} \left[ y_j \log \sigma \left( \frac{\psi_i(x_j) + b}{s} \right) + (1 - y_j) \log \left( 1 - \sigma \left( \frac{\psi_i(x_j) + b}{s} \right) \right) \right]$$
(16)

where pseudo-labels $y_j$ are assigned as: $y_j = \begin{cases} 1 & \text{if } x_j = x_{k_i}^* \text{ and } \delta_i = 1, \\ 0 & \text{otherwise.} \end{cases}$

Consequently, the collaborative surrogate loss: $\mathcal{L}_{\text{surrogate*}} = \mathcal{L}_{\text{CMPLL}} + \lambda \mathcal{L}_{\text{CBCE*}}$

### 5.2.1 GRADIENT UPDATES: GLOBAL LEARNING

During training, $\mathcal{L}_{surrogate*}$ propagates gradients to the agent $i$ through the update:

$$\hat{\theta}^i \leftarrow \hat{\theta}^i - \eta \left( \nabla_{\hat{\theta}^i} \mathcal{L}_{\text{CMPLL}} + \lambda \nabla_{\hat{\theta}^i} \mathcal{L}_{\text{CBCE*}} \right)$$
(17)

In equation 15 and equation 16, the loss is a function of both $\hat{\theta}^i$ and the parameters $\hat{\theta}^j$ of neighbouring agents $j \in \mathcal{N}_i$. Consequently, gradient updates must also propagate to all $j \in \mathcal{N}_i$, ensuring collaborative refinement of preference estimates across the population.

We adopt a two-phased training framework analogous to meta-learning. First a "base learner" updates individual agent parameters $\hat{\theta}^i$ in equation 17. Then a "global learner" enforces consistency across neighbourhoods via regularisation. After updating $\hat{\theta}^i$ through Equation 17, neighbouring agents $j \in \mathcal{N}_i$ are updated upon incorporating a regularisation term $R_{\hat{\theta}^i} = \sum_{k \in \mathcal{N}_i} d\left(\hat{\theta}^i, \hat{\theta}^k\right)$:

$$\hat{\theta}^j \leftarrow \hat{\theta}^j - \eta \left(\nabla_{\hat{\theta}^j} \mathcal{L}_{\text{CMPLL}} + \lambda \nabla_{\hat{\theta}^j} \mathcal{L}_{\text{BCE}} + \beta \nabla_{\hat{\theta}^j} R_{\hat{\theta}^i}\right) \quad \forall j \in \mathcal{N}_i \tag{18}$$

This framework allows a single data point $t \in \mathcal{D}_{\mathcal{U}}$ to influence multiple agents $j \in \mathcal{N}_i$, reducing per-agent data requirements to train a large population of agents. We empirically investigate the quantitative savings in data requirements in our experiments.

**Cold Start Problem:** The collaborative training model addresses the cold-start problem by initialising a new agent $\theta$ only after they provide a data point. For new users without data, predictions can made using a global preference vector (e.g. the weighted average of $\hat{\Theta}$). Over a population of large users, this approach leads to significantly less data requirements, which consequently improves the user-experience as an agent is required to provide minimal data for a reasonable level of accuracy. We explore this hyper-sparse data availability scenario in detail in our experiments[2]. The training algorithm is described in Appendix E and Figure 1 provides a high level overview of the model architecture.

## 6 EXPERIMENTS

We design our experiments to validate core contributions of our framework: **1) inferring user preferences from sparse binary feedback**, and **2) improving data efficiency through collaborative training**. Our evaluation proceeds in three stages:

1. **Framework Analysis**: We first use a simulated dataset to analyse our weakly supervised DFL approach in a controlled setting, measuring decision quality against a fully supervised oracle and performing ablation studies.
2. **Real-World Application**: We the evaluate our framework on a real-world hyper-sparse trip-planning dataset to demonstrate its effectiveness in a challenging, real-world environment.
3. **Standardised Benchmarking**: Finally, we benchmark our collaborative training mechanism against state-of-the-art models on the MovieLens-1M (24) cold-start recommendation task (25) to validate its generalisability and robustness.

### 6.1 ANALYSIS OF THE WEAKLY SUPERVISED DFL APPROACH

We evaluate our framework on a simulated dataset with access to ground-truth user parameters $\theta$ to report exact decision regret. Table 1 summarises the core results of our model with key ablations. A Prediction-Focused Model with linear predictor was chosen as the baseline for our experiments. Detailed model selection, ablation study, sparsity analyses, and additional diagnostics (including $P_5$ statistics and nDCG results) are discussed in Appendix D.1.4.

Table 1: Analysis of decision quality and data efficiency on the simulated dataset (mean $\pm$ std)

| Model | Normalised Regret (5 Samples) | Mean Accuracy (2 Samples) | Mean Accuracy (5 Samples) |
|---|---|---|---|
| Baseline (PFL) | $4.17 \pm 0.69$ | $79.6\% \pm 0.1$ | $90.7\% \pm 0.9$ |
| CT-only | $3.07 \pm 1.26$ | $66.6\% \pm 0.2$ | $74.4\% \pm 0.2$ |
| DFL-only | $0.40 \pm 0.21$ | $92.7\% \pm 0.1$ | $97.4\% \pm 0.1$ |
| **DFL + CT (Full Model)** | $\mathbf{0.21 \pm 0.14}$ | $\mathbf{97.9\% \pm 0.3}$ | $\mathbf{100.0\% \pm 0.0}$ |

---

[2]The experiments in this work were performed on an M1 ARM processor with 16GB unified RAM.

The results show two clear findings. First, the decision-focused DFL objective is essential for minimising regret: the Baseline and CT-only models incur high normalised regret (4.17 and 3.07), while DFL-only reduces regret to 0.40 and the full DFL+CT model further improves to 0.21. Second, the effect of collaborative training (CT) is nuanced. In isolation, the CT-only model's predictive accuracy is lower than the baseline's; however, it still achieves a lower decision regret. This indicates that even in a zero-shot setting—trained solely on neighbour signals —the CT mechanism can infer a meaningful preference structure, despite weaker predictive performance.

## 6.2 Personalised Trip Planning

We next evaluate our framework on the motivating application of personalised trip planning, using a real-world feedback dataset with 20k anonymised interactions from 12k users. The extreme sparsity of this dataset —with a median of only one interaction per user ($91\%$ of users have $\leq 2$ interactions)—makes preference learning highly challenging. Since the dataset does not contain ground-truth parameters ($\theta$ or $x^*(\theta)$), we cannot compute true regret and instead report accuracy and ROC-AUC, summarised with ablation results in Table 2. Models trained without a decision-focused objective (Baseline, CT-only) perform poorly, with CT-only collapsing to near-random accuracy. Introducing the DFL loss drastically improves performance, and the full **DFL+CT** model achieves the best results with $78.3\%$ accuracy and an ROC-AUC of $83.5\%$, representing a gain of over 23 percentage points compared to the strongest baseline.

To verify that our collaborative training mechanism captures latent preference structure, we project the learned user embeddings ($\hat{\theta}^i$) into two dimensions using t-SNE. As shown in Figure 2, distinct clusters emerge despite all embeddings being initialised at zero, providing qualitative evidence that the model discovers meaningful user groupings. Users with very sparse interactions appear in the middle of the projection but are oriented toward clusters, suggesting that even limited data guides them toward preference groups through the collaborative signal.

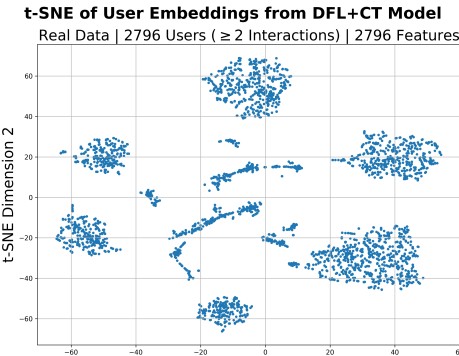

Table 2: Ablation study on real feedback data (mean $\pm$ std)

| Model | Accuracy (%) | ROC-AUC (%) |
|---|---|---|
| Baseline (PFL) | $54.72 \pm 0.96$ | $55.57 \pm 0.50$ |
| DFL-only | $71.66 \pm 0.35$ | $78.28 \pm 0.42$ |
| CT-only | $33.93 \pm 0.44$ | $50.17 \pm 0.12$ |
| w/o MPLL | $67.62 \pm 0.43$ | $50.25 \pm 0.12$ |
| w/o PseudoLabels | $77.11 \pm 0.40$ | $63.08 \pm 0.16$ |
| PseudoLabels-Top3 | $47.94 \pm 0.57$ | $61.26 \pm 0.43$ |
| **DFL+CT (Full Model)** | $78.28 \pm 0.29$ | $83.49 \pm 0.34$ |

Figure 2: t-SNE visualisation of learned user embeddings from the trip-planning dataset, reflecting distinct latent preference clusters.

## 6.3 Benchmarking collaborative training on cold-start recommendation

The goal of this experiment is not to propose a new state-of-the-art recommender system, but to validate that our general-purpose, meta-learning framework for handling data sparsity is competitive with specialised architectures. We follow a user-level cold-start protocol on MovieLens-1M (24), treating users below the 80th percentile of interaction counts as "new". For each new user, we split their data into 20% support, 10% validation and 70% query sets, a more challenging split than the 70/10/20 used in literature. We choose a simple linear model as the predictor in our DFL framework.

As shown in Table 3, our simple Linear+DF+CT model achieves the highest ranking quality, with $nDCG@5 = 0.799$. This strong ranking performance, achieved despite a higher MSE, is a direct result of our surrogate loss 11, which can be tuned via $\lambda$ to prioritise ranking quality over prediction error, a critical feature in recommendation scenarios.

Table 3: Cold-Start Recommendation Performance Comparison

| Model | Data Split Ratio (Training:Validation:Testing) | MSE | MAE | nDCG@5 |
|---|---|---|---|---|
| **PAML*** | $7 : 1 : 2$ | $1.210 \pm 0.029$ | NA | $0.779 \pm 0.002$ |
| **Wide and Deep** | $2 : 1 : 7$ | $1.109 \pm 0.043$ | $0.835 \pm 0.015$ | $0.700 \pm 0.015$ |
| **DropoutNet** | $2 : 1 : 7$ | $\mathbf{0.969 \pm 0.013}$ | $\mathbf{0.787 \pm 0.011}$ | $0.725 \pm 0.004$ |
| **MeLU** | $2 : 1 : 7$ | $1.122 \pm 0.043$ | $0.838 \pm 0.015$ | $0.699 \pm 0.008$ |
| **MetaCS** | $2 : 1 : 7$ | $2.990 \pm 0.021$ | $1.353 \pm 0.005$ | $0.733 \pm 0.002$ |
| **DFL + CT (Ours)** | $2 : 1 : 7$ | $1.110 \pm 0.002$ | $0.844 \pm 0.001$ | $\mathbf{0.799 \pm 0.001}$ |

\* *Metrics shown for this model are as observed by Yu (8) for a 7:1:2 data split.*

## 7 CONCLUSION

This work addresses the challenge of deploying a Decision-Focused Learning (DFL) system in real-world scenarios where ground-truth data is expensive or impractical to collect. Combining DFL with meta-learning, we present a framework that overcomes two key obstacles: weak supervision from binary feedback and data sparsity in cold-start settings. Our results demonstrate that it is indeed possible to learn user preferences via non-invasive feedback as supervision without compromising on performance. The proposed surrogate decision loss seem to work well and the training algorithm is easy to implement. We also demonstrate the potential of meta-learning to improve the performance of DFL in data sparse settings. We apply our framework to the personalised trip planning problem and the results show a potential of our model in improving public transit accessibility.

**LLM Usage**  The authors acknowledge the use of a large language model (LLM) for editorial support in refining the language and structure of this paper. All scientific contributions, methodologies, and conclusions presented are solely the work of the authors.

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

## A EXTENDED RELATED WORKS

Decision-Focused Learning (DFL), also known as predict-and-optimise, trains predictive models to directly optimise downstream decision tasks. Early work in this area proposed, for example, using continuous relaxations to differentiate through discrete solvers (11) to train the model for decision quality rather than parameter prediction accuracy. Wilder et al. (2) introduced a general DFL framework for combinatorial problems, showing that directly optimising the decision objective often outperforms standard two-stage pipelines. Elmachtoub and Grigas (4) formalised this idea with the "Smart Predict-Then-Optimize" (SPO) framework, which trains predictors by minimising a surrogate of the induced decision error when predicted cost parameters are used in a linear (or integer) optimisation. Other works since have derived specialised losses or algorithms for DFL, from differentiating through interior-point LP solvers (12) to learning-to-rank (7) and locally optimised decision losses (6).

Mandi et al. (1), in their foundational DFL work, note that the problem of predicting the parameters of the optimisation problem is typically a supervised ML problem and the targets may either be observed as ground truth parameters or optimal decision values. In practical application, however, one may lack explicit labels; instead, only an aggregate or binary reward signal (e.g. success/failure of the chosen solution) may be available. Mulamba et al. (3) hint at this setting by treating non-optimal solutions as "negative examples". However, such approach still assume having access to a correlated dataset of ground truth parameters as target variables.

Despite extensive research and growing interest in DFL under full supervision, settings involving sparse weakly-supervised binary feedback signals on decision outcomes remain largely unexplored. Although related areas (such as reinforcement learning or bandit optimisation) tackle decision making with limited feedback (13; 26; 14; 15), they do not generally consider the decision quality of the downstream combinatorial problems.

This work is motivated by the personalised trip-planning problem in public transit within a multi-agent framework. The problem can be viewed as a specialised case of personalised recommendation, which has been studied extensively (9; 27–32). The key distinction, however, is that trip planning requires users to interact with the optimisation problem of finding an optimal path parametrised by their individual preferences. Traditional approaches apply deep learning (16), reinforcement learning (17), or manual preference surveys (18). Recent work has explored framing recommendation as a decision-focused learning problem (see Wilder et al. (2) for an example applied to diverse recommendation), demonstrating the feasibility of a DFL formulation for recommendation tasks. Those approaches, however, typically operate under full supervision (with observed problem parameters or explicit labels) whereas our work tackles a more challenging scenario of sparse, weak binary feedback on decision outcomes.

We address this gap by remodelling the decision-focused objective as a weakly supervised DFL problem for settings in which, instead of ground-truth parameters or optimal decisions, the learner receives only sparse binary outcome signals (e.g. success/failure) for chosen solutions.

## B MOTIVATING EXAMPLE: DFL VS. STANDARD CLASSIFICATION

In this example, we demonstrate howing that our model drives $\theta$ towards optimal decision-making instead of picking the highest rated items like a standard classifier.

**Setup.** Consider a path-finding scenario where options are defined by two features: $x = (\text{time}, \text{cost})$. A user's true (but unobserved) utility parameter is $\theta^* = [-1, 0]$, meaning they want to **minimise time** and are completely **indifferent to cost**. The available data consists of two queries (see Table 4):

Table 4: Motivating Example - DFL

| Query | Options | Feedback |
|-------|---------|----------|
| Q1 | $x_1 = (7,6),\ x_2 = (12,1),\ x_3 = (11,5)$ | $f(x_1, \theta^*) = 1$ |
| Q2 | $x_2 = (12,1),\ x_4 = (9,5),\ x_5 = (15,3)$ | $f(x_2, \theta^*) = 0$ |

**Baseline: Standard Classifier (BCE Loss).** A standard classifier or recommender system only sees *two* item-feedback pairs, i.e. the available dataset $D = \{(x_1, 1), (x_2, 0)\}$. It is trained to predict the feedback, typically by minimizing Binary Cross-Entropy (BCE). With an initial model parameter $\theta_0 = [0, 0]$ and assuming a sigmoid prediction $\sigma(\theta^\top x)$, the initial gradients for each query are:

- Q1 Gradient: $[0.5 - 1, 0] \cdot x_1 = [-3.5, -3]$
- Q2 Gradient: $[0.5 - 0, 0] \cdot x_2 = [6, 0.5]$

The total gradient is $\nabla_{\text{BCE}} = [-3.5, -3] + [6, 0.5] = [2.5, -2.5]$. A gradient step $\theta_1 \leftarrow \theta_0 - \eta \nabla_{\text{BCE}}$ will push the 'time' component of $\theta$ negative (correct) but will also push the 'cost' component positive (incorrect). The model is incorrectly learning to favour high cost because it correlates with acceptable time in the limited data.

**Our Approach: DFL Surrogate Loss ($L_{\text{MPLL}} + L_{\text{BCE}^*}$).** Our method uses the full context of the decision problem. We define the surrogate loss (Section 11) as

$$L_{sr} = L_{MPLL} + L_{BCE}$$

where $\lambda = 1$ and $L_{MPLL}$ and $L_{BCE^*}$ are defined in equation 9 and 10 respectively. Their corresponding gradients are:

$$\nabla L_{MPLL} = -(2\delta_i - 1) \sum_{j \neq i^*} \sigma(-\gamma_{ij}) \left[ u_i^*(1 - u_i^*) x_i^* - u_j(1 - u_j) x_j \right]$$

$$\nabla L_{BCE^*} = \sum_j (\sigma(\theta^\top x) - y_j) x_j$$

At $\theta_0 = [0, 0]$, we first generate pseudo-labels for all options to get the dataset:

We first assign psuedo-labels to get the dataset in Table **??**

| Query | (Psuedo-) Labelled Dataset |
|---|---|
| Q1 | $\{(x_1, 1), (x_2, 0), (x_3, 0)\}$ |
| Q2 | $\{(x_2, 0), (x_4, 0), (x_5, 0)\}$ |

The gradients are then computed using our full surrogate loss:

- Q1 Gradients: $\nabla_{\text{MPLL}} = [1.125, -0.75]$, $\nabla_{\text{BCE}^*} = [2.66, 0]$
- Q2 Gradients: $\nabla_{\text{MPLL}} = [0, -0.75]$, $\nabla_{\text{BCE}^*} = [6, 1.5]$

The total gradient is $\nabla_{\text{sur}} = [1.125, -0.75] + [2.66, 0] + [0, -0.75] + [6, 1.5] = [9.785, 0]$. A gradient step $\theta_1 \leftarrow \theta_0 - \eta \nabla_{\text{sur}}$ pushes the 'time' component of $\theta$ negative (correct) while leaving the 'cost' component unchanged (correct).

**Conclusion.** The baseline gradient $\nabla_{\text{BCE}} = [2.5, -2.5]$ or even pseudo-label-enhanced gradient $\nabla(BCE^*) = [4.33, 0.75]$ learns an incorrect correlation, leading to suboptimal decisions. Our DFL gradient $\nabla_{\text{sur}} = [9.785, 0]$ correctly identifies that only time matters for the decision, perfectly aligning with the true utility $\theta^* = [-1, 0]$ and demonstrating a truly decision-focused learning process.

## C  RELATING ACCEPTABILITY PROBABILITY TO EXPECTED REGRET

This section shows that maximising the probability of producing an "acceptable" decision is a valid surrogate for minimising expected regret: increasing the acceptability probability tightens a natural upper bound on expected regret. Expectation below is taken over the data distribution $(z, \theta) \sim \mathcal{D}$ and any predictor randomness.

**Proposition 1.** *Let $(z, \theta)$ be a data instance, where $z$ are features and $\theta$ are the true parameters. For a decision $x$ define the regret*

$$\text{Regret}(\theta, x) := u(x^*(\theta), \theta) - u(x, \theta) \geq 0.$$

*Assume there exist finite constants $U_{\max} \geq 0$ and $\Delta_{\max} \geq 0$ such that almost surely*

$$0 \leq u(x^*(\theta), \theta) \leq U_{\max}, \qquad 0 \leq \text{Regret}(\theta, x) \leq \Delta_{\max}.$$

*Fix a tolerance $\tau \in (0, 1]$ and define the acceptability indicator*

$$f = \mathbf{1}\{ u(x, \theta) \geq \tau \cdot u(x^*(\theta), \theta) \}.$$

*Then the expected regret satisfies the uniform upper bound*

$$\mathbb{E}[\text{Regret}(\theta, x)] \leq \Delta_{\max} - (\Delta_{\max} - (1 - \tau)U_{\max}) P(f = 1). \tag{19}$$

*In particular, if $\Delta_{\max} > (1 - \tau)U_{\max}$, increasing $P(f = 1)$ strictly tightens the upper bound on expected regret equation 19.*

*Proof.* Consider the two possible values of $f$.

If $f = 1$ then $u(x, \theta) \geq \tau\, u(x^*(\theta), \theta)$, so

$$\text{Regret}(\theta, x) = u(x^*(\theta), \theta) - u(x, \theta) \leq (1 - \tau)\, u(x^*(\theta), \theta).$$

If $f = 0$ we use the trivial bound $\text{Regret}(\theta, x) \leq \Delta_{\max}$. Combining these two outcomes gives the exact pointwise inequality

$$\text{Regret}(\theta, x) \leq (1 - \tau)\, u(x^*(\theta), \theta)\, f + \Delta_{\max}(1 - f).$$

Applying the uniform bound $u(x^*(\theta), \theta) \leq U_{\max}$ yields

$$\text{Regret}(\theta, x) \leq (1 - \tau)U_{\max}\, f + \Delta_{\max}(1 - f) = \Delta_{\max} - (\Delta_{\max} - (1 - \tau)U_{\max})\, f,$$

and taking expectation over $(z, \theta)$ (and any model randomness) gives equation 19. $\qquad\square$

**Illustration.** Suppose $U_{\max} = 100$, $\Delta_{\max} = 50$, and $\tau = 0.9$ (i.e., acceptable means achieving at least 90% of the optimum). Then $(1 - \tau)U_{\max} = 10$ and $\Delta_{\max} - (1 - \tau)U_{\max} = 40$, so equation 19 gives

$$\mathbb{E}[\text{Regret}] \leq 50 - 40 \cdot P(f = 1).$$

A model with $P(f = 1) = 0.2$ has expected-regret upper bound $50 - 40 \cdot 0.2 = 42$, whereas a model with $P(f = 1) = 0.8$ has upper bound $50 - 40 \cdot 0.8 = 18$. The instance-aware bound equation **??** can be substantially tighter if, for example, optimal utilities on the $f = 1$ subset are much smaller than the uniform $U_{\max}$.

**Remarks.**

- **One-sidedness and looseness.** The bound is *one-sided*: increasing $P(f = 1)$ guarantees a decrease of the *upper bound* on expected regret, but does not tightly control regret in the $f = 0$ ("unacceptable") region.
- **Non-differentiability and training.** The indicator $f$ is not differentiable, so directly maximizing $P(f = 1)$ is not suitable for gradient-based learning. In practice, one may use a differentiable surrogate (e.g. a smooth approximation to the indicator, or a probabilistic model followed by a likelihood / cross-entropy training objective). Our surrogate loss in provides one such differentiable signal while also furnishing gradients that help discriminate different failing cases for $f = 0$.

# D    EXPERIMENT DETAILS

In this section, we provide a detailed description of the experimental setup along with an analyses of the datasets used in our work.

## D.1 TRIP PLANNING EXPERIMENT

### D.1.1 INTERACTION FRAMEWORK

The experimental framework operates on a dataset $\mathcal{D} = (\mathcal{U}, \mathcal{Q}, \mathcal{F})$ and a supplementary knowledge base $\mathcal{P}$ where:

- $\mathcal{U}$ denotes the set of users
- $\mathcal{Q} \subset \mathcal{S} \times \mathcal{S}$ represents query pairs from the search space $\mathcal{S}$
- $\mathcal{P} = \bigcup_{q \in \mathcal{Q}} \mathcal{P}(q)$ constitutes the collection of all possible paths, where $\mathcal{P}(q)$ denotes the path set for query $q$
- $\mathcal{F} : \mathcal{U} \times \mathcal{Q} \times \mathcal{P} \to \{0, 1\}$ represents the binary feedback function

For each query $q \in \mathcal{Q}$, there exists a non-empty path set $\mathcal{P}(q) \subseteq \mathcal{P}$. User interactions follow a sparse observation paradigm where each user $u \in \mathcal{U}$ engages with a subset $\mathcal{Q}_u \subset \mathcal{Q}$ such that $|\mathcal{Q}_u| = N_u \ll |\mathcal{Q}|$.

Each interaction instance corresponds to a tuple $(u, q, p, P(q), f)$ where:

- $u \in \mathcal{U}$ denotes the interacting user
- $q \in \mathcal{Q}_u$ specifies the query pair
- $p \in \mathcal{P}(q)$ identifies the evaluated path
- $\mathcal{P}(q)$ describes the path set for the query
- $f \in \{0, 1\}$ represents the user's binary feedback

Note that, the observational model only records feedback for the path $p$ presented to the user $u$, leaving the hypothetical feedback for alternative paths $\mathcal{P}(q) \setminus \{p\}$ unobserved.

### D.1.2 FEEDBACK DATASET

Our real-world feedback dataset consists of $18,210$ binary evaluations from $12,180$ users from a trip planner deployed in a metropolitan transit network, recording over four months on a network of approximately $7,000$ bus stops and $285$ metro stations. Prior to analysis, we applied standard cleaning; excluding zero-distance queries, excluding trip-time outliers and the top 1% most frequent origin-destination pairs.

Despite its overall size, the per-user feedback is highly sparse: the median user contributes one query, and over 91% of users supply at most two feedback instances. Fewer than ten users exceed 30 queries (omitted from Fig. 3a for clarity). Spatially, interactions span some 3842 distinct $0.1° \times 0.1°$ grid cells, though the busiest cell—near the City Centre —accounts for 21% of all queries (Fig. 3b). Temporally, query rates fall below $100\,h^{-1}$ before $05:00$, rise to a morning peak of approximately $1180\,h^{-1}$ ($08:00 - 10:00$), remain elevated through midday, and reach a secondary maximum of around $1430\,h^{-1}$ at $18:00$ (Fig. 3d).

The data also exhibits significant class imbalance: Overall positive feedback is 68.2%, but declines to roughly 55% for users with six or more queries (Fig. 3d). This extreme per-user sparsity —given the overall volume —of the dataset motivated our decision-focused, meta-learning approach.

### D.1.3 SIMULATED DATA GENERATION

Simulated feedback was generated using a population of users and paths designed to produce predictable interaction patterns. The dataset was intentionally structured to be linearly separable in high-dimensional space to validate the model's fundamental operation: collaborative training using shared feedback and learning latent user preference mappings.

Each user $u \in \mathcal{U}$ is represented as a real vector in $\mathbb{R}^n$, where $n$ denotes the dimensionality of the search space. Experiments were conducted with $n = 2$ for visual interpretability and $n = 6$ to evaluate performance in higher-dimensional regimes.

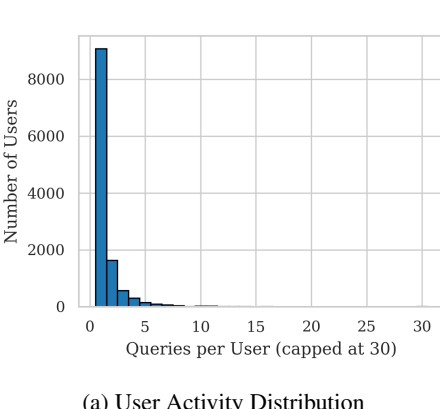

(a) User Activity Distribution

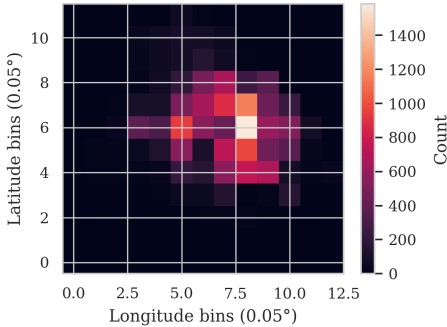

(b) Origin Heatmap (Coordinates Removed From Axes Labels for Anonymity)

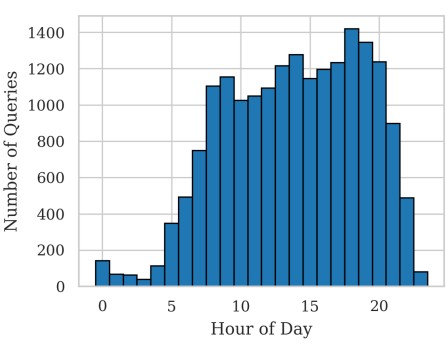

(c) Temporal Query Pattern

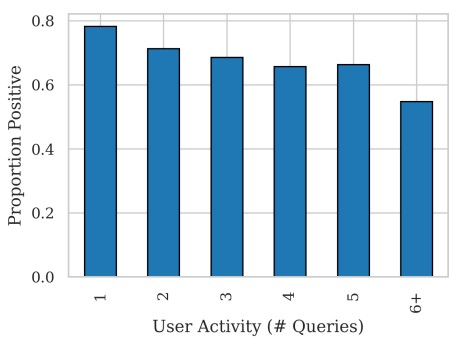

(d) Feedback vs. Activity

Figure 3: Real Feedback Data Description

For an $n$-dimensional search space, both users and paths are partitioned into $n$ mutually exclusive preference groups. The user population decomposes as $\mathcal{U} = \bigcup_{i=1}^{n} \mathcal{U}_i$ where $\mathcal{U}_i \cap \mathcal{U}_j = \emptyset$ for $i \neq j$. This property does not hold for path set $\mathcal{P} = \bigcup_{j=1}^{m} \mathcal{P}_j$

User group $\mathcal{U}_i$ is characterized by activation in the $i^{th}$ dimension: For $u^i \in \mathcal{U}_i$,

$$u^i[k] \sim \begin{cases} Gamma(c, \sigma) & k = i \\ Gamma(\epsilon, \sigma) & k \neq i \end{cases}$$

where $c > 0$ and $\epsilon = 0.1$ controls non-preferred feature weights.

Correspondingly, paths $p^i \in \mathcal{P}_i$ exhibit complementary structure:

$$p^i[k] \sim \begin{cases} Gamma(c, \sigma) & k = i \\ -Gamma(c, \sigma) & k \neq i \end{cases}$$

Feedback generation follows an affinity principle: For interaction $(u, p)$,

$$f = \begin{cases} 1 & \text{if } u \in \mathcal{U}_i \text{ and } p \in \mathcal{P}_i \\ 0 & \text{otherwise} \end{cases}$$

This structure ensures $u \cdot p > 0$ for aligned groups and $u \cdot p < 0$ otherwise by design, maintaining linear separability through opposing sign patterns in non-preferred dimensions.

### D.1.4 EXPERIMENTAL SETUP

The experimental framework evaluates a Decision-Focused Learning with Collaborative Training (DFL+CT) architecture for personalised feedback prediction. We construct a simulated dataset containing $N_{\text{users}}$ users, each associated with $N_{\text{interactions}}$ interaction tuples $(u, q, p, P(q), f)$, where $P(q)$ denotes path properties and $f \in \{0, 1\}$ represents binary feedback.

**Validation Protocol** We employ user-specific leave-$k$-out cross-validation with three training regimes defined by per-user interaction counts $N_{\text{train}} \in \{2, 3, 5\}$. For each user, $N_{\text{train}}$ interactions are randomly selected for training while the remaining $N_{\text{interactions}} - N_{\text{train}}$ form the test set.

**Model Selection** For evaluation of the feedback dataset, we extended our simulated framework through two methodological adaptations to address practical challenges mentioned in Section D.1.2. First, we choose a prediction-focused-learning model as a baseline: the 'predictor' is independently trained for minimising feedback prediction error. Upon convergence, the parameters from the predictor ('weights') are used to solve the optimisation problem. Secondly, to select the best model for the predictor, we evaluate performance between standard classifier models (see Figure 4) and choose the best performing one. Additionally, the DFL+CT implementation utilises a linear predictor with the Collaborative MPLL (CMPLL) (Section 5.2).

**Optimisation Details** We conduct hyperparameter optimisation through 100 Optuna trials (33): For the synthetic data, the objective is to minimise normalised regret, whereas for the real feedback data, is is to maximise the validation area under the receiver operating characteristic curve (ROC-AUC). Final results report mean performance metrics with $\pm$ standard deviation error bounds across five independent runs using optimal hyperparameters.

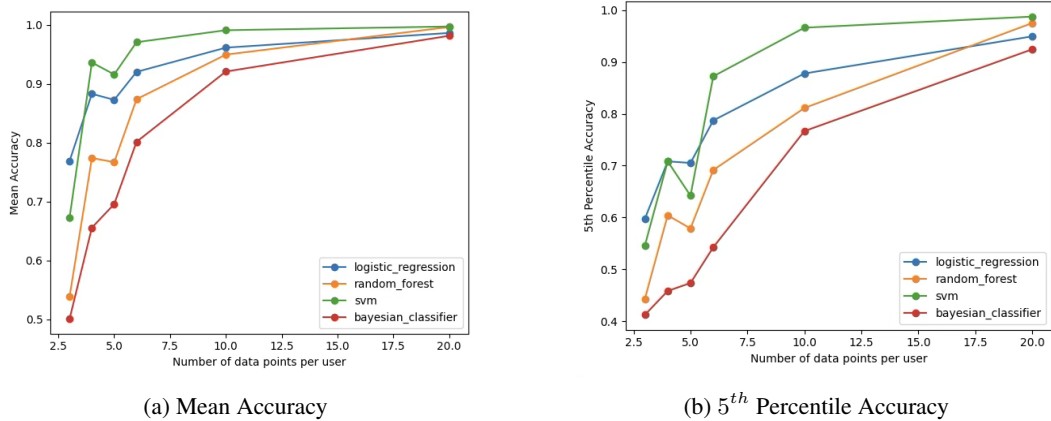

(a) Mean Accuracy

(b) $5^{th}$ Percentile Accuracy

Figure 4: Performance of Baseline Models for Simulated Data

**Simulated Data:** Figure 4a plots the mean prediction accuracy of all four classifier configurations as a function of the number of training interactions per user. As expected, accuracy improves monotonically for every method as $N_{\text{train}}$ increases. Among the off-the-shelf baselines, support vector machines attain the highest mean accuracy in the low-data regime ($\approx 68\%$ at $N_{\text{train}} = 3$), while the Bayesian classifier performs worst ($\approx 65\%$ at $N_{\text{train}} = 5$). All baselines approach near-perfect accuracy when $N_{\text{train}} = 20$.

Figure 4b shows the 5th-percentile accuracy, highlighting performance variability across users. Here the baselines exhibit substantial worst-case degradation at small $N_{\text{train}}$: for example, the Bayesian classifier's $P_5$ accuracy is only 41% at $N_{\text{train}} = 3$, and random forest falls below 60% at $N_{\text{train}} = 5$.

**Real Feedback Data:** The evaluation of our models on real feedback data (Table 2) validates the patterns observed in simulated experiments, with baseline models failing to generalise or performing much worse than DFL and CT+DFL models, and CT+DFL model outperforming all models. Specifically, the CT+DFL architecture achieves the best performance with 78.28% test accuracy and 83.48% ROC-AUC, representing improvements of 22.98 and 26.48 percentage points respectively over the strongest baseline (XGBoost).

**Visual Analysis:** Figure 5 illustrates a cold-start scenario for a single user in a two-dimensional setting. We assume a population of 200 users, already clustered into two groups, and introduce one new user who contributes ten data points. To simulate labelling, we select one existing user at

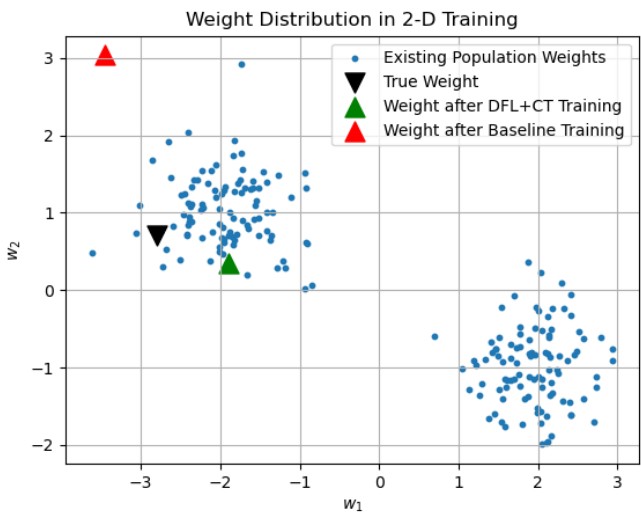

Figure 5: Cold-Start for a User in 2D

random to act as the "true user" and use their responses to generate the target values. Each new data point is fed to the true user to obtain labels, which are then used to train both the models.

As before, we compare our model (DFL + CT + baseline) directly against the baseline; in this case the baseline is an SVM model. Our combined DFL + CT approach consistently learns weight vectors that lie close to those of the user's nearest population neighbours—and therefore close to the true user's weight. Although in this simple 2-D example the SVM baseline can score slightly higher on standard evaluation metrics (because it sits further from the artificial linear boundary between clusters), this separation-agnostic behaviour can be detrimental when clusters are not linearly separable or in high-dimensional feature spaces, as observed through the decision quality (regret) performance in our experimental results (Section 6). In such cases, the ability of our framework to align with the latent preference characteristics leads to substantially better performance, demonstrating its capacity to recover true user preferences.

Overall, the proposed CT+DFL architecture not only boosts average predictive performance but also dramatically improves worst-case accuracy in sparse-data regimes in addition to generalising to imbalanced class distribution, demonstrating its effectiveness for personalised feedback prediction.

**Ablation Studies**

COMPLETE ABLATION STUDY RESULTS    This section provides the complete results of the ablation study on the simulated dataset, as summarised in the main text. The study was designed to dissect the contribution of each component of our framework and surrogate loss function. The results are presented in Table 5.

The findings confirm that the full DFL+CT framework yields the best decision quality, evidenced by the lowest normalised regret. The removal of the DFL objective ('CT-only') leads to a catastrophic decline in decision quality (a > 15x increase in regret), confirming that a decision-focused objective is essential to the validity of the framework.

The analysis of the surrogate loss components also reveals important insights. The pseudo-labelled BCE component is critical; its removal ('w/o BCE Pseudo-labels') results in a sevenfold increase in regret. Conversely, removing the pairwise ranking loss ('w/o MPLL') has a minimal impact on regret, suggesting its primary role is to fine-tune ranking metrics rather than drive core decision quality. Notably, an alternative pseudo-labelling strategy ('Pseudo-labels @Top3') improves the ranking metric nDCG@5 to the highest of any configuration, but at the cost of a fivefold increase in regret which is not worthwhile trade-off.

Table 5: Complete ablation study results on the simulated dataset. The full framework achieves the best decision quality (lowest normalised regret).

| Model Configuration | nDCG@5 | Normalised Regret |
|---|---|---|
| DFL + CT (Full Framework) | 0.9134 | $0.2059 \pm 0.1406$ |
| DFL-only | 0.9021 | $0.3984 \pm 0.2060$ |
| CT-only | 0.9058 | $3.0738 \pm 1.2580$ |
| w/o MPLL | 0.9096 | $0.2199 \pm 0.1428$ |
| w/o BCE Pseudo-labels | 0.9021 | $1.4545 \pm 0.6758$ |
| Pseudo-labels @Top3 | 0.9171 | $1.0393 \pm 0.2409$ |

COMPLETE RESULTS FOR DATA SPARSITY EXPERIMENT    Here we present the complete results for the data sparsity experiment on the simulated dataset, including the 5th percentile ($P_5$) analysis, which reflects the worst-case performance across users. The results, shown in Table 6, demonstrate the data efficiency and robustness of our framework.

As noted in the main text, the DFL+CT framework achieves near-perfect mean accuracy with only five data points per user. The $P_5$ analysis further strengthens this finding, showing that our framework provides a high performance floor even in the most challenging, low-data scenarios. For example, with only two data points, the baseline classifier's worst-case accuracy is poor (60.6%), whereas our full framework maintains a robust $P_5$ accuracy of 91.7%. This confirms that the collaborative training mechanism not only improves average performance but also makes the model significantly more reliable for all users.

Table 6: Complete performance comparison on the simulated dataset, showing Mean and 5th percentile ($P_5$) accuracy as a function of the number of training data points per user.

| Data Points/User | Baseline Classifier | | DFL-only | | DFL+CT (Ours) | |
|---|---|---|---|---|---|---|
| | Mean Acc. | $P_5$ Acc. | Mean Acc. | $P_5$ Acc. | Mean Acc. | $P_5$ Acc. |
| 2 | $79.6 \pm 0.1$ | $60.6 \pm 1.7$ | $92.7 \pm 0.1$ | $84.8 \pm 0.2$ | $97.9 \pm 0.3$ | $91.7 \pm 3.9$ |
| 3 | $84.4 \pm 2.0$ | $67.3 \pm 2.2$ | $95.2 \pm 0.4$ | $87.6 \pm 0.9$ | $99.4 \pm 0.1$ | $95.9 \pm 0.5$ |
| 5 | $90.7 \pm 0.9$ | $76.1 \pm 1.7$ | $97.4 \pm 0.1$ | $92.2 \pm 0.1$ | $100.0 \pm 0.0$ | $100.0 \pm 0.0$ |

Table 7: Ablation Study Setup

| Ablation | Description |
|---|---|
| DFL+CT | Full model run |
| DFL only | Each user is independently trained with the surrogate loss in equation 11 only. No collaborative training between users |
| CT only | Gradient updates are done solely through collaborative loss in equation 15. |
| w/o MPLL | The pairwise logistic loss is not included, the surrogate loss is only the modified BCE Loss |
| w/o PsuedoLabels | No psuedo labels are generated. The BCE Loss component only evaluates the provided feedback |
| PseudoLabels - TopK | Top K options in terms of predicted utility are assigned a positive feedback instead of all pseudo labels being negative |

## D.2  COLD START RECOMMENDATION EXPERIMENT

### D.2.1  DATA DESCRIPTION

To evaluate our cold-start approach in a more conventional recommendation setting, we use the MovieLens-1M dataset (24). After standard preprocessing, the dataset consists of 1000209 explicit

ratings (on a 1-5 scale) from 6040 users on 3952 movies. Following common practice in implicit-feedback evaluations, we binarise the data by treating ratings of 4 or 5 as positive interactions and all others as negatives.

### D.2.2 EXPERIMENTAL SETUP

We simulate the cold-start scenario at the user level by identifying "new" users as those whose total number of positive interactions falls below the 80th percentile of the number of user interactions, in line with prior work on meta-learning for recommendation (8). In contrast to the conventional interaction-level split of 70% train/10% validation/20% test (34), we adopt a more challenging 20%/10%/70% split for each cold user:

- **Support set (20%)**: a very small number of interactions used to adapt the meta-learner to a new user.
- **Validation set (10%)**: held-out interactions for early-stop and hyperparameter tuning.
- **Query set (70%)**: the remaining interactions used to evaluate recommendation quality under cold-start.

We repeat this sampling procedure five times with different random seeds and report the average performance (± standard deviation).

Our base recommendation model is a simple linear predictor (i.e., user and item embeddings combined via a dot-product and linear transformation), identical to that used in our trip-planning experiments. Although our cold-start optimisation is agnostic to the base model architecture, we chose this linear form to isolate the gains attributable solely to the meta-learning adaptation. We measure accuracy using MSE, MAE errors on the held-out query set and Normalised Discounted Cumulative Gain@5 (nDCG@5) to capture ranking performance in the cold-start regime.

## E  TRAINING ALGORITHM

Algorithm 1 describes the training algorithm used in our framework through stochastic gradient descent. As the 'label' of every data point is a user-generated feedback, every data point is associated with a particular user. In every iteration, the 'neighbourhood' of the user in question is calculated. This is to ensure that user preferences are characterised based on the total information available at any given time. While a cluster of users might exhibit similar preferences, as more data points become available, every vector is refined, and the preferences might change.

In addition to usual hyper-parameters like learning rate ($\eta_1, \eta_2$) and regularisation factor ($\rho$), three crucial parameters affect the performance of the model. Firstly, the parameter $\alpha$ describes the magnitude of the impact the predictions of the neighbourhood have on the prediction of a user. Secondly, the parameter $\epsilon$ represents how 'big' the neighbourhood is defined to be. For example, in Euclidean terms, $\epsilon$ describes the radius of a sphere around a user such that every user within the sphere lies in the same neighbourhood. A small constant $\epsilon_0$ is to avoid the self assignment of a user to its neighbourhood[3]. Finally, the $\lambda$ parameter sets the trade-off between the ranking quality and prediction accuracy (Section 4.1.1).

### E.1  PERFORMANCE ANALYSIS

#### E.1.1  RUNTIME ANALYSIS

Assigning agents to user-specific neighbourhoods as described in Section 5 entails a $\mathcal{O}(n^2)$ pairwise distance evaluations. While these computations can be vectorised, the dominant cost arises from computing the collaborative utility equation 13 and corresponding gradient updates. We scale the radius $\epsilon$ by a factor of $(1 + \gamma)$ at intervals of $\eta$ iterations.

To assess the impact of these scheduling parameters on the per-epoch wall-clock time $T$, we assume a multiplicative (power-law) relationship,

$$T = C \, \eta^p \, \gamma^q,$$

---

[3]We define a neighbourhood of a user to be the association of other users to it.

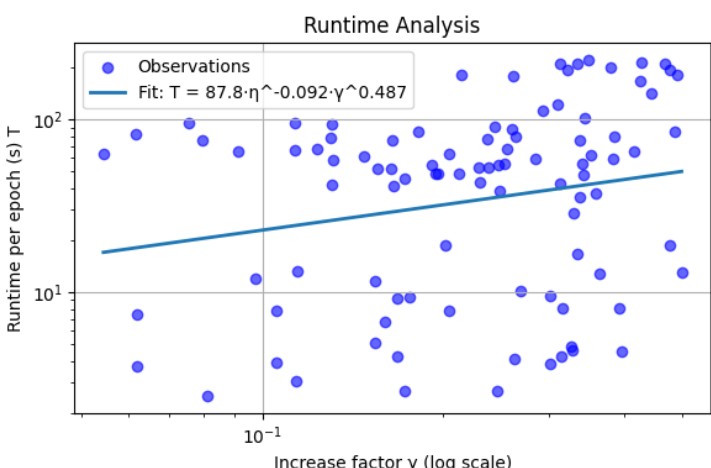

Figure 6: Runtime Complexity Analysis.

which is equivalently expressed in logarithmic form as

$$\log T = \log C + p \log \eta + q \log \gamma$$

This transformation permits estimation of the coefficients $(C, p, q)$ by ordinary least squares. Fitting this model across 100 Optuna trials results in $T \approx 87.8 \ \eta^{-0.092} \ \gamma^{0.487}$. The exponent $q = 0.487$ (Figure 6) indicates that doubling $\gamma$ increases per-epoch runtime by approximately 40%, while $p = -0.092$ reveals a modest reduction in runtime when updates are applied less frequently. Consequently, $\gamma$ may represent a special target for further optimisation.

### E.1.2 HYPERPARAMETER ROBUSTNESS ANALYSIS

We conducted a robustness analysis to evaluate the model's sensitivity to the collaborative training hyperparameters: the neighbourhood threshold ($\epsilon$) and the collaborative weight ($\alpha$). We used 122 distinct $(\epsilon, \alpha)$ pairs from our Optuna runs, with $\alpha \in [0.10, 0.84]$ and $\epsilon \in [0.11, 4.00]$. We quantify robustness using a sensitivity-based metric:

$$\textbf{Robustness Score} = \frac{1}{1 + \textbf{Sensitivity Ratio}}$$

where **Sensitivity Ratio** $= \dfrac{\text{CV(Performance)}}{\text{CV(Parameter)}}$, and CV denotes the coefficient of variation. This score ranges from 0 to 1 (higher is more robust) and was measured using Accuracy and ROC-AUC.

**Collaborative Weight ($\alpha$) Analysis**   Using both Pearson and Spearman analysis, we find that $\alpha$ exhibits moderate statistically significant negative correlations with both Accuracy ($r = -0.357, p < 0.001$) and ROC-AUC ($r = -0.347, p < 0.001$). This reflects the expected over-smoothing effect at high $\alpha$ values: with an increase in $\alpha$, the predicted utility largely depends on the "neighbourhood" consensus rather than user-preference. Despite this, the model achieves high robustness scores of 0.71 (Accuracy) and 0.78 (ROC-AUC), confirming stable performance across most of its effective range.

**Neighbourhood Threshold ($\epsilon$) Analysis**   The neighbourhood threshold $\epsilon$ shows weaker correlations with performance (Accuracy $r = -0.120$, ROC-AUC $r = -0.237$), with only the correlation for ROC-AUC being statistically significant ($p = 0.009$). Importantly, $\epsilon$ demonstrates superior robustness scores of 0.79 and 0.85, indicating the model's high stability across different neighbourhood sizes.

**Conclusion**   The overall analysis yields an aggregate robustness score of 0.78, classifying our DFL with Collaborative Training (DFL+CT) framework as highly robust. All correlation magnitudes

are below 0.5, indicating no strong dependencies on specific hyperparameter values. The neighbourhood threshold ($\epsilon$) exhibits greater robustness than the collaborative weight ($\alpha$), suggesting the neighbourhood-based learning approach is inherently stable. The observed negative correlation with $\alpha$ is within theoretical expectations and does not compromise model stability.

---

**Algorithm 1** Training Algorithm

**Input:**

- A set of users: $i \in \mathcal{U}$.
- A global dataset $\mathcal{D}_{\mathcal{U}}$, represented as:

$$\mathcal{D}_{\mathcal{U}} = \left\{ \left( i, z_{ji}, x_{ji}^*, y_{ji} \right) \,\Big|\, i \in \mathcal{U}, \ j \in \{1, \ldots, |\mathcal{D}_i|\} \right\}$$

  where:
  - $x_{ji}^*(\theta')$: A suggested optimal solution to user $i$.
  - $y_{ji} = f(x_{ji}^*, \theta^i)$: Feedback from user $i$ on the suggested solution.
- A global decision space mapping $\mathcal{X} : z \to \mathcal{X}_z$
- Parameters:
  - Learning rates: $\eta_1$ (base learner), $\eta_2$ (global learner).
  - Hyperparameters: $\alpha$ (collaboration weight), $\lambda$ (loss tradeoff), $\rho$ (regularization weight)
  - Distance thresholds: $\epsilon_o, \epsilon$ (neighbourhood limits).
  - Classification margin $b \in R$

**Output:**

- Trained model parameters $\hat{\theta}^i$ for each user $i \in \mathcal{U}$.

**Initialisation:**

- $\hat{\Theta} = \{\}$: Dictionary to store encountered users and their parameters.
- $random\_initialiser()$: Random weight initialiser.
- $\mathcal{N}_i = \emptyset$: Set of neighbourhood for all $i$.

1: **Training Loop**
2: **for** each epoch **do**
3:     **for** each $(i, z_{ji}, x_{ji}^*(\theta'), y_{ji}) \in \mathcal{D}_{\mathcal{U}}$ **do**
4:         **if** $i \notin \hat{\Theta}$ **then**                    ▷ New user encountered
5:             $\hat{c}_i \leftarrow random\_initialiser()$
6:             $\hat{\Theta}[i] \leftarrow \hat{\theta}^i$
7:         **else**                              ▷ Retrieve parameters for existing user
8:             $\hat{\theta}^i \leftarrow \hat{\Theta}[i]$
9:         **end if**
10:
11:         $\mathcal{N}_i \leftarrow \left\{ j \mid j \in \hat{\Theta}, \epsilon_o \leq d(\hat{\theta}^i, \hat{\theta}^j) \leq \epsilon \right\}$          ▷ Identify Neighbourhood
12:         $\mathcal{X}_{z_{ji}} \leftarrow \mathcal{X}(z_{ji})$                       ▷ Get Decision Space
13:
14:         $\psi_i(x) = (1-\alpha)\theta^{i\top} x + \alpha \cdot \dfrac{\sum_{j \in \mathcal{N}_i} \frac{\theta^{j\top} x}{d(\theta^i, \theta^j)}}{\sum_{j \in \mathcal{N}_i} \frac{1}{d(\theta^i, \theta^j)}}$    ▷ Latent Utility Score
15:
16:         $u(x, \theta^i) = \sigma\left( \frac{\psi_i(x) + b}{s} \right)$              ▷ Collaborative Utility Score
17:
18:         $\mathcal{L}_{surrogate*} \leftarrow \mathcal{L}_{CMPLL} + \lambda \mathcal{L}_{CBCE*}$       ▷ Calculate Surrogate Loss
19:         $\hat{\theta}^i \leftarrow \hat{\theta}^i - \eta_1 \nabla_{\hat{\theta}^i} \mathcal{L}_{surrogate*}$
20:
21:         $R_{\hat{\theta}^i} \leftarrow \sum_{j \in \mathcal{N}_i} d(\hat{\theta}^i, \hat{\theta}^j)$            ▷ Regularisation Parameter
22:
23:         $\mathcal{L}_{meta} \leftarrow \mathcal{L}_{surrogate*} + \rho R_{\hat{\theta}^i}$         ▷ Global Learner Update
24:         **for** $j \in \mathcal{N}_i$ **do**
25:             $\hat{\theta}^j \leftarrow \hat{\theta}^j - \eta_2 \nabla_{\hat{\theta}^j} \mathcal{L}_{meta}$
26:         **end for**
27:     **end for**
28: **end for**

