# OpenReview forum: "Towards Decision Focused Learning for Sparse and Weakly Supervised Environments"
_ICLR.cc/2026/Conference — Submitted to ICLR 2026_

### Official Review · Reviewer_pMCb · 2025-10-27

**Soundness:** 3
**Presentation:** 2
**Contribution:** 2
**Rating:** 2
**Confidence:** 3

**Summary:**

This paper extends the decision-focused learning framework to scenarios involving sparse, binary supervision signals within a multi-agent system. In particular, the authors focus on recommendation tasks aimed at predicting user preferences. Based on several assumptions and approximation adaptations, they design the MPLL loss and BCE loss within a collaborative training and meta-learning framework for optimization. Experiments are conducted on both synthetic data and a trip-planning recommendation dataset to show the effectiveness of the proposed model.

**Strengths:**

1. The problem under study is novel.
2. Mathematical analyses are provided to illustrate the proposed methodology.
3. The ablation study demonstrates the effectiveness of the proposed model components.

**Weaknesses:**

1. Generally, I feel that the intuition and benefits of the proposed method are not clearly articulated. There exists a substantial body of prior work on personalization/recommendation with sparse supervised learning signals. It is therefore unclear why decision-focused learning is necessary for this problem and what specific advantages it offers. The authors introduce many modifications based on various assumptions for adaptation, and the resulting optimization algorithm appears considerably more complex than standard supervised learning approaches. Without a clear intuitive justification, this added complexity seems difficult to motivate.
2. Some concepts lack clear and practical definitions. For example, it is not explained what the solution variable  $x$ repsent in real recommender system, or why the utility function can be defined as the sigmoid of a linear combination of the solution and the parameters being optimized. The papar says "DFL trains the predictor so that the predicted parameters leads to an optimal output from the solver", yet it remains unclear what the solver specifically refers to and why it is necessary.
3. The introduction of the multi-agent setting further adds confusion. The motivation and necessity of incorporating this into the decision-focused learning framework are not well justified. It is not evident what benefits or practical implications this multi-agent design brings to recommender systems.
4. The practical implementation of these concepts in the experiments is not clearly described. Moreover, since the code and data are not released, it is difficult for other researchers to understand, verify, or reproduce the results.

**Questions:**

My main questions are outlined in the weaknesses section. They primarily concern the benefits and necessity of the proposed method, as well as the practical meaning of its underlying concepts.

---

> ### Author Response · Authors · 2025-11-19
> **Response to Reviewer pMCb (pt 1/2)**
>
> We thank the reviewer for their remarks and would like to clarify and answer their questions in our two part response.
>
> **Weakness 1**
>
> 1) *Generally, I feel that the intuition and benefits of the proposed method are not clearly articulated. There exists a substantial body of prior work on personalization/recommendation with sparse supervised learning signals.*
>
> The focus of our work is not on personalisation/ recommendation using sparse supervised learning signals. Instead, we work on decision-focused learning (DFL) framework for scenarios where the only available target is weakly supervised binary feedback based on decision outcomes (see Section 1, lines 47-48).
>
> In our work, we present two experiments to demonstrate the application of DFL: Personalised Trip Planning and Cold Start Recommendation. However, DFL is also applicable in Linear Optimisation, Knapsack, Travelling Salesman Problem, Portfolio Optimisation, Energy Scheduling and several other problems [1]. Our primary contribution is to extend traditional DFL to weak supervision: rather than training on datasets like $\\{(z_i,\theta_i)\\}$, we consider a setting where neither $\theta$ nor the corresponding optimal solution $x^\*(\theta)$ is observed. Instead, we only receive a sparse binary evaluation $f(x,\theta)$ of a decision $x$, generated using some initial parameter $\theta'$ (See Section 3.1 lines 149-153).
>
> The primary experiment discussed in our work on DFL involves personalised trip planning (see Section 1, lines 87-89), a real-world application of the shortest path problem, where we learn the optimisation parameters for each user based on their sparse binary feedback. DFL is the appropriate framework to find the shortest path when the “weights” of the edges in a graph are unknown but can be predicted using a correlated dataset, say, historical edge travel times. DFL trains a prediction model so that predicted travel times produce shortest paths that are close to those computed with true costs. In our setting, no historical costs are available; we only observe whether a computed path was “short enough” for a user and infer each user’s optimisation parameter from these observations. To our knowledge, this is the first attempt to frame a DFL problem in a weakly supervised setting.We provide theoretical and empirical evidence (Section 6.1; Appendix B) that DFL outperforms Prediction-Focused Learning (PFL); which first predicts the parameter $\theta$ before solving the optimisation problem.
>
> The second experiment, which involves cold-start recommendation, is indeed extensively explored in the literature. However, as noted in Section 1 (line 92) and Section 6.3 (lines 477-479), our goal with this experiment is not to claim state-of-the-art in cold start recommendation, but to demonstrate the generalisability of the collaborative training framework.
>
> 2) *It is therefore unclear why decision-focused learning is necessary for this problem and what specific advantages it offers.*
>
> The primary difference between Decision Focused Learning (DFL) and recommendation systems is that DFL operates in a fundamentally different output space than standard recommendation systems. In a typical recommender system, the model selects from a *finite, shared catalogue of items*, such as a fixed list of movies or products. DFL, by contrast, generates or selects decisions from a **dynamic and potentially infinite solution space** defined by an optimisation problem (Section 1). For example, in personalised trip-planning, the optimal route for a user’s trip is computed by solving an optimisation problem (e.g. shortest path with personalised cost): The goal of DFL is to estimate the cost such that the computed shortest path is closest to the actual shortest path. Methods like collaborative filtering or matrix factorisation, assume an underlying structure of users and items with historical ratings and learn latent factors to generalise preferences within that fixed universe. DFL, on the other hand, learns to produce optimal decisions by integrating predictive modelling with an optimisation problem. Essentially every instance of an optimisation problem generates a new user-item matrix and there are almost no feedback available for each of the items. The feedback in our setting is only available on the decision quality. In other words, a recommender system cannot be directly applied to this setting while DFL can be used for recommendation system[2] (Section 6.2). In summary, DFL is not a variant of a recommendation ranking task, but a different class of problem.
>
> [1] J. Mandi et al., ‘Decision-Focused Learning: Foundations, State of the Art, Benchmark and Future Opportunities’, J. Artif. Int. Res., vol. 80, Sept. 2024, doi: 10.1613/jair.1.15320.
>
> [2] B. Wilder, B. Dilkina, and M. Tambe, ‘Melding the Data-Decisions Pipeline: Decision-Focused Learning for Combinatorial Optimization’, AAAI, vol. 33, no. 01, pp. 1658–1665, July 2019, doi: 10.1609/aaai.v33i01.33011658.

---

> ### Author Response · Authors · 2025-11-19
> **Response to Reviewer pMCb (pt 2/2)**
>
> **Weakness 2:**
>
> We would like to answer this question in three parts:
> 1. The cold-start recommendation experiment conducted in this study follows the standard implementation protocol used in recommendation systems. Relevant works on data splitting and the experimental setup are cited in Section 6, line 413, with detailed descriptions and explanations provided in Appendix D.2. Similarly, the mathematical model for the trip planning experiment is described in detail in Appendix D.1, with the appendix being referenced in Section 6.1, line 422.
> 2. The motivation for modelling the utility function as a sigmoid has been discussed in detail in Section 3.1 and formalised in Section 3.2. Since we do not have access to the actual ground-truth parameter, denoted as $\theta$, we approximate the objective function by maximising the probability of receiving positive feedback. Using a sigmoid function for this purpose is the simplest choice, as it simplifies the analysis and description of loss functions and the overall framework (Section 3.2 lines 185-190).
> 3. By "solver," we refer to any tool (e.g. combinatorial solver) used to solve the optimisation problem (Section 1, line 70). As mentioned previously, Decision-focused learning (DFL), by definition, involves a combination of prediction and optimisation models (Section 1, lines 36-46). The main goal of a DFL framework is to solve an optimisation problem of the form $\arg\max u(x, \theta)$, where $\theta$ is a parameter of the objective function and is unknown (Section 3 lines 126-136). To account for the absence of $\theta$, a prediction model $m(\omega, z)$ is trained using a correlated dataset $\{(z_i, \theta_i)\}_{i = 1}^N$ (Section 3.1 lines 147). A prediction $\hat\theta = m(\omega, z)$ is then used to obtain a predicted solution $x^\*(\hat\theta)$ by solving the optimisation problem $\arg\max u(x, \hat\theta)$ through any relevant (e.g. CO) solver. This ``solver" is what we refer to in the text.
>
> **Weakness 3**
>
> To reiterate, the primary goal of our work is not to design recommender systems. We state this in Section 1, line 88, as well as in Section 6.3, line 451. We chose to use a recommendation system as a demonstrative experiment to validate our meta-learning (collaborative training) framework's ability to handle data sparsity (see Section 6.3, lines 452-453).
>
> Regarding our motivation for employing a multi-agent setting, we explain in Section 1, lines 57-60, that one of the challenges in datasets with binary feedback is that a single agent may not generate sufficient data for meaningful learning. Consequently, in scenarios where hyper-personalised decision-making is crucial for each agent, we introduce an additional collaborative training mechanism to address this issue of data sparsity. Thus, while our feedback-based DFL framework can function independently for each agent when enough data is available, we also provide a mechanism to tackle data scarcity.
>
> However, as the other reviewer noted, this distinction was not entirely clear. We have clarified in Section 3.1 that we first describe the model for a single agent before expanding to the multi-agent setting.
>
> **Weakness 4**
>
> The primary practical application of our work lies in scenarios where the parameters necessary to solve an optimisation problem are unknown and need to be estimated. Instead of having a comprehensive dataset, we only possess a weakly labelled feedback dataset derived from previous decisions. For example, consider the problem of stocking a warehouse. The objective is to stock items that yield the highest revenue, denoted as $x^\*(\theta) = \arg\max u(x, \theta)$, but the expected revenue for each item ($\theta$) is unknown.
>
> In traditional decision-focused learning (DFL), this problem is addressed by using a predictive model based on a correlated dataset, such as historical data on item prices $(\{(z_i, \theta_i)\}_{i=1}^N)$. This data is used to estimate item revenues ($\hat\theta$), allowing the selection of items for stocking that minimises revenue loss compared to the true optimal choice $(x^\*(\theta))$.
>
> In our scenario, however, we do not have access to historical data. Instead, we have binary feedback indicating whether some selected set of items $(x^\*)$ generated sufficient revenue or not $(f(x, \theta))$, without knowing the actual revenue of the items in that lead to that choice ($\theta$). Our work focuses on solving the DFL problem using this weakly supervised dataset.
>
> Thank you for pointing out the absence of code and data. We have included the code and (raw and pre-processed) dataset used for our experiments in the uploaded revision.

---

### Official Review · Reviewer_4C8D · 2025-10-30

**Soundness:** 3
**Presentation:** 3
**Contribution:** 3
**Rating:** 6
**Confidence:** 2

**Summary:**

This paper tackles decision-focused learning (DFL) when only sparse binary feedback (“acceptable” vs “not”) is available and many users have just 1–2 interactions. It proposes an end-to-end surrogate that optimizes the quality of chosen decisions rather than predictive accuracy, combining a modified pairwise logistic loss with a conservative pseudo-label BCE so that the model learns to rank the selected solution above (or below) alternatives depending on the feedback. To handle cold start, it adds a collaborative layer: each user’s score blends their own preference vector with a distance-weighted consensus of neighbors, followed by two-phase updates that adjust both the focal user and their neighbors. The framework is validated on a personalized trip-planning task and a user-level cold-start benchmark (MovieLens-1M), where the simple Linear+DFL+Collaborative variant achieves state-of-the-art ranking (e.g., nDCG@5 ≈ 0.799) despite higher MSE, aligning training with the decision objective.

**Strengths:**

The method aligns training directly with the downstream decision objective by optimizing whether the actually chosen option is acceptable, rather than predicting proxy scores; it turns a single binary feedback into rich learning signal via a modified pairwise ranking loss combined with a conservative pseudo-label BCE, yielding data efficiency under sparse supervision.

**Weaknesses:**

### Robustness and Ablations
Evidence comes mainly from one trip-planning setup and a linear MovieLens variant, so generality is unclear. Reported gains may hinge on specific design choices (surrogate mix, consensus rule, data curation). Does the effect persist with other model classes, alternative decision-focused surrogates, and consensus mechanisms? How sensitive are results to neighborhood size, distance and other hyperparameters?

### Scalability Concerns
Neighbor search and consensus updates can dominate runtime and memory as users/items/dimensions grow. System behavior under load spikes and rolling updates is not characterized. What are the time/memory scaling laws at large scale? What is the end-to-end latency budget (including neighborhood rebuild frequency) for online inference and updates?

**Questions:**

See weaknesses

---

> ### Author Response · Authors · 2025-11-24
> **Response to Reviewer 4C8D**
>
> We thank the reviewer for an encouraging and a thorough review, and on questions regarding framework generality and design choices. We have addressed these points below:
>
> # Robustness and Ablations:
>  **Generality:** We first emphasise that our work is the first to explore the application of Decision Focused Learning (DFL) when only sparse binary feedback on decision outcomes is available; this has been wholly unexplored in recent literature [1], but has numerous potential applications. Our goal through this work is to formalise this setting and provide a few examples where such setup can be utilised. Our setup can indeed be applied to a large class of problems where the decision is made by solving a combinatorial optimisation problem. However, we refrain from making such claims at this time. Having said that, post-submission simulations on other problems (such as linear optimisation) suggest a broader generality of our method, including the surrogate loss functions that we propose in our work. Indeed, this could be a future research direction in the weakly supervised DFL setup.
>
> Additionally, the personalised trip-planning problem presented in our work is also the first application of DFL to trip-planning in public transit systems, and our work uses a real-world dataset collected from public transit users to demonstrate the feasibility of our solution. We believe that a personalised trip planning algorithm that is motivated by DFL is itself a challenging and novel problem. However, to demonstrate generality beyond this domain:
> 1. **Theoretical Support:** In Prop. 1 (Appendix C), we show that maximising the probability of generating "acceptable" solutions (positive feedback) tightens the upper bound on expected regret. This establishes a general reduction for DFL under binary supervision: the intractable goal of minimising regret without ground truth can be replaced by the tractable goal of maximising acceptance probability.
> 2. **Additional Experiments** Since submission, we expanded our evaluation to include optimisation problems of varied scale, such as the Knapsack and general Linear Programming (LP). Our preliminary results indicate that our approach converges to minimal regret here as well. While the regret is naturally higher than fully supervised DFL (as we approximate ground truth), it remains sufficiently close to the global minima. Code for these additional experiments is included in the supplementary material.
>
> **Robustness of Design Choices** Regarding whether dependence on design choices, we argue that our architecture is necessitated by intrinsic data constraints:
>
> 1. **Surrogate Loss:** Standard DFL surrogate losses require ground-truth parameters or optimal decisions, which are unavailable in our setting. Our surrogate design (combining ranking (MPLL) and prediction (BCE*)) results from having only binary feedback.
> 2. **Ablations:** We performed detailed ablation studies (Table 2, 5, and Appendix D.1.4 lines 1009-1025) to validate our surrogate loss. Results show that removing either the ranking (MPLL) or prediction component (Pseudo-label BCE) degrades performance.
> 3. **Persistence across Model Classes:** While the dual-component structure of the surrogate loss is critical, the specific functions can vary. For instance, in our LP experiments (see supplementary material), we successfully replaced the sigmoid utility ($\sigma(\theta^\top x)$) and BCE with a simple logit ($\theta^\top x$) and Hinge loss. This suggests that the framework is flexible with respect to the specific loss functions used.
>
> **Hyperparameter Sensitivity** The performance of our model primarily depends on two hyperparameters: neighbourhood threshold $\epsilon$ and collaborative weight $\alpha$. We address the sensitivity in Appendix E.1.2 and show that our model is highly stable across these parameters.
>
> # Scalability Concerns:
>
>  The reviewer correctly points out that the neighbourhood search and consensus updates are the main factors contributing to potential slowdowns. We address these issues via:
> 1. **Controlled $\epsilon$:** We limit computational cost by defining the neighbourhood using radius $\epsilon$ (Section 5, line 300). This parameter is scaled dynamically by a factor of $(1 + \gamma)$ at regular intervals of $\eta$ iterations to manage the trade-off between collaboration and run-time (Appendix E.1.1).
> 2. **Empirical Scaling Laws:** As detailed in our runtime analysis (Appendix E.1.1), we modelled the per-epoch wall-clock time $T$ as a power-law function: $T \approx 87.8 \eta^{-0.092} \gamma^{0.487}$. This analysis reveals that doubling the scaling factor $\gamma$ increases runtime by approximately 40\%, while the update frequency $\eta$ has a minimal impact, suggesting the system is robust to variations in update scheduling.
>
> [1] J. Mandi et al., ‘Decision-Focused Learning: Foundations, State of the Art, Benchmark and Future Opportunities’, J. Artif. Int. Res., vol. 80, Sept. 2024, doi: 10.1613/jair.1.15320.

---

### Official Review · Reviewer_rNw4 · 2025-10-31

**Soundness:** 3
**Presentation:** 2
**Contribution:** 2
**Rating:** 2
**Confidence:** 3

**Summary:**

This paper proposes a DFL framework for settings with only binary feedback, learning latent user preferences through a differentiable surrogate loss and meta-learning mechanism. Experiments show that it significantly reduces decision regret and performs well in data-sparse and cold-start scenarios.

**Strengths:**

1. The problem of using only binary feedback to solve DFL is interesting and important, which could be a more practical avenue for DFL.

2. It also studies the collaborative training, in which the data from users with similar preferences are shared. Ablation studies show that this collaborative training mechanism helps to get a better performance.

**Weaknesses:**

1. There are many typos and incorrect paragraphs in the paper. For example, the use of the notation $u(x, \theta)$ is very confusing. In Section 3.1, $u(\theta)$ is introduced as a general function and $f$ is defined by Eq. (3).  However, in Section 3.2, Eq. (5) provides $u(x,\theta) = P(f(x,\theta)=1)$. These two uses are not compatible if Eqs. (3) and (5) are taken together.  It is also unclear what the randomness in $P(f(x,\theta)=1\mid x,\theta)$ refers to. If the $x, \theta$ is given, the function $f(x,\theta)$ should be deterministic. The typos and incorrect equations could take much effort to understand the true meaning of this paragraph.

2. Also, the assumption in Section 4.1 is very weird and hard to satisfy. Since $f(x^\*, \theta)$ can either be 0 or 1, this assumption implies that for any $x^\*$, either $u(x^\*, \hat{\theta}) \ge u(x_j, \hat{\theta})$ for all $x_j \in X_z\setminus \\{x^\*\\}$, or  $u(x^\*, \hat{\theta}) \ge u(x_j, \hat{\theta})$ for all $x_j \in X_z\setminus \\{x^\*\\}$. This is a very strict assumption. I understand that this is not a true “assumption” but rather a motivation for introducing the modified Pairwise Logistic loss and BCE loss. It would be clearer to present this assumption as a motivation instead of labeling it as an assumption.

**Questions:**

1. Could the author explain why the loss minimization is related to the regret minimization? Is there any theoretical result that can show the relationship between these two goals?

2. In order to achieve collaborative training, the algorithm needs to update $\hat{\theta}$ at each round and try to find the neighbors at each time. Is it too costly in time? In the experiment, does the author have any tricks to avoid finding neighbors for each gradient update?

---

> ### Author Response · Authors · 2025-11-19
>
> We thank the reviewer for their helpful and informative review. Below we address the reviewers' comments.
>
> **Weakness 1:** We thank the reviewer for highlighting the confusion around the notation $u(x,\theta)$ and its relationship to $\Pr[f(x,\theta)=1 \mid x,\theta]$.
>
> In Section 3.1, we introduce the \emph{general} Decision-Focused Learning (DFL) framework in abstract form. There, $u(x,\theta)$ is presented as a generic objective (utility) function of an optimisation problem:
>
> $$x^*(\theta) \in \arg\max_{x \in \mathcal{X}} u(x,\theta).$$
>
> The parameters $\theta$ are unknown, and the goal in DFL is to learn $\theta$ so that the resulting decision $x^*(\theta)$ has low regret, as formalised in Eq.~(2). We also describe the standard supervised DFL setting where a dataset $D=\\{(z_i,\theta_i)\\}_{i=1}^N$ is available and a prediction model $m(\omega,z)$ is trained so that $\hat{\theta}=m(\omega,z)$ minimises regret. In contrast, our work considers a \emph{weakly supervised} setting, where the dataset consists of triples
>
> $$
> D=\\{(z_i, x_i^*, f_i)\\}_{i=1}^N,
> $$
>
> with $x_i^\*  = x^\*(\theta')$ generated from an initial estimate $\theta'$, and $f_i = f(x_i^\*,\theta_i)$ is binary feedback under the (latent) true parameter $\theta_i$. Thus, instead of observing $\theta_i$, we observe only a sparse and noisy abstraction of decision quality via $f(x,\theta)$, as described in Eq. (3).
>
> Section 3.2 then *instantiates* the abstract DFL objective for this weak-feedback scenario. Since our observations are binary, we define
>
> $$
> u(x,\theta) := \Pr[f(x,\theta)=1 \mid x,\theta],
> $$
>
> which is Eq. (5). This is a *specific* instantiation suitable for our setting and not intended to conflict with the general definition in Eq. (1).
>
> Regarding the source of randomness in $\Pr[f(x,\theta)=1 \mid x,\theta]$:  while $f(x,\theta)$ would be deterministic under known $\theta$ with no noise, in our setting $\theta$ is latent and feedback is modelled as stochastic to capture noise and unobserved variability. This is noted in Section~3.1 (lines 149--153, 167--170). Additionally, we only have access to feedback on a single decision variable, $x \in \mathcal{X}$, within the entire decision space. We reiterate that the goal of a Decision Focused Learning problem is to estimate $\theta$ which leads to minimal regret for the downstream optimisation problem. The definition in Equation (5) is consistent with this goal.
>
> We finally want to point out to the reviewer that setting proposed in the paper is  novel and there are no prior works in the literature that we could find. Additional post-submission experiments on other categories of optimisation problems (like linear) further support the generality of the approach, though we do not add new claims to the paper.
>
> **Weakness 2:** The reviewer accurately notes that the assumption in Section 4.1 is indeed strict but one that serves as the basis for the loss functions described in our work. Our intention was to use it as a *motivating idealisation* for the construction of our pairwise logistic and BCE losses, rather than as a realistic condition that must always hold.
>
> Given that $f(x,\theta)$ is a weak, noisy abstraction of the latent parameter $\theta$, and that only a single decision $x^\*(\theta')$ is observed per instance, such a simplifying condition provides useful guidance in designing losses that still promote learning parameters $\theta^*\$ yielding low regret.
>
> Empirically, both the experiments in the submission and additional post-submission tests indicate that the framework performs robustly even when this condition is only approximately satisfied. **In the revision, we have relabeled this assumption explicitly as a *motivating simplification* and clarify its practical role.**
>
> **Question 1:** We examine this relationship in Appendix C, where we show that maximising the probability of producing an `` acceptable" decision is a valid surrogate for minimising expected regret: increasing the acceptability probability tightens a natural upper bound on expected regret.  The reviewer may want to read Proposition 1 for a  theoretical argument.  Subsequently, the connection between pairwise logistic loss and binary cross-entropy (BCE) loss with respect to maximising the probability of positive outcome is trivial.
>
> **Question 2:** The calculation of neighbours can be resource-intensive but our experiments suggest it is not costly in time; It's overall impact on computation is considerably lessened by the neighbourhood radius parameter, $\epsilon$, which restricts the search area (see Section 5 line 307). This parameter is further scaled by a factor of $(1 + \gamma)$ at intervals of $\eta$ iterations; our primary optimisation strategy to prevent a global search for each upgrade ( runtime analysis in Appendix E.1.1). Although more efficient techniques may exist, our experiments indicated no considerable slowdown or necessity to explore additional optimisations at our current scale.

---

### Official Review · Reviewer_rr4b · 2025-11-01

**Soundness:** 2
**Presentation:** 2
**Contribution:** 3
**Rating:** 4
**Confidence:** 3

**Summary:**

This paper tries to address an important problem for decision-focused learning. Prior DFL methods assume access to rich supervision information, which is not realistic in many real-world settings where only sparse, weak feedback is available. This paper proposed a new framework that DFL can work even when only sparse and binary feedback is available, which is common in many real-world decision systems. Decision regret and ranking quality, as the performance metric, make more sense than the prediction accuracy.

**Strengths:**

- The identified problem is important, and the writing of the paper describes it in a natural tone.
- Have novel theoretical results and well-rounded analysis.
- The design of the surrogate loss is the highlight of the paper.

**Weaknesses:**

- The model and setup are still confusing in different dimensions.
- Experiments are not sufficient to address or verify the claim.
Please refer to the details in the Questions.

I will consider raising my score after the rebuttal questions have been properly addressed. But I do think this paper requires quite a lot of editing and polishing to reach the publishable stage.

**Questions:**

Literature:
- The related works seem not to cover enough literature. Although I'm not very familiar with the DFL/PFL framework, this reminds me of a few terms, such as Smart Predict-then-Optimize (PTO), Joint Prediction and Optimization, and end-to-end learning. Would you like to compare and contrast these related literature? Is there any effort from these works in order to address noise and limited feedback/labels?
- In your setup, you have a latent assumption on the tolerance of decision makers that is a key structure for getting your theoretical results, thus you may also want to mention this compromising/tolerance effect from other literature, such as psychological/social sciences, to support and justify your setup.

Model:
- You mentioned "multi-agent" setting in your introduction, but the parameter $\theta^*$ is assumed to be fixed for all agents. How does the "multi-agent" make the problem more interesting/complex? In other words, what makes it necessary to introduce the multi-agent environment?
**Add-ons:** I read Section 5, and this question has been partially addressed. But I do recommend introducing multi-agent at the beginning and mentioning that you first start with a single agent.
- What are the assumptions of the utility function $u(x,\theta)$ (e.g. shape of $u$ w.r.t. $x$ and $\theta$, contuity)?
- In equation (2), I did not quite get it. Is the $\theta$ the ground-truth parameter? I feel it is quite confusing because $\theta$ is usually a variable, and $\theta^*$ is usually the optimal value of parameters. Afterwards, in line 141-142, you also mention $\theta_i$ as the ground-truth and I become confused -- are there $N$ different ground truths? **typo**: should be $\{\dots\}_{i=1}^N$, and $N$ is not defined.
- Gradually following your description of the model, I sort of understand this is an online learning setup, but it is not stated clearly in the introduction. Until you define the regret, I have not realized that this is not an offline joint optimization and prediction problem. You may want to clarify somewhere at the beginning.
- Line 147-148: $f(x, \theta)$ is defined as a function w.r.t. both $x$ and $\theta$. Unless you assume $\theta$ is the ground truth, you need to define what is $\theta$, otherwise, you may not include $\theta$ as a parameter of $f$.
- As you introduced below equation (2), the surrogate loss is to address minimizing the regret in non-trivial cases. Then in line 153 and below, you mention it is the unknown $\theta$ that makes the regret intractable. This makes the purpose of the introduction of the surrogate loss a bit misleading. Does your setup also consider the discrete action set $\mathcal{X}$? Does the surrogate loss help both in terms of the $\arg\max$ and the unknown $\theta$?
- In line 174-179, I am not so convinced the choice of sigmoidal function is the right nonlinear mapping to be chosen. I understand that it may provide a nice structure for the proof, but it would be good to support your choice by citing other literature or mentioning that the empirical performance is sufficient. You may want to check/compare with other mapping empirically. **typo**:  $s,b$ are not defined. I'm confused whether these noise & scaling are tuning hyperparameters.
- A follow-up question: Is the method sensitive to the design of the surrogate loss or hyperparameters?
- in line 197 & 206, multiple $\mathcal{X}_{?}$ are not properly defined.
- The resolution of Figure 1 needs to be improved. Consider outputting PDF format instead.
- seems to Section 5.0.1 is a wrong index?
- Can the approach be extended beyond binary feedback to other objective settings (e.g. ordinal, multiple target)?

Experiments:
- For your experiments in Section 6.1, I don't think you can produce meaningful conclusions with merely 5 data samples/users. Would you provide more explanation of why you think this suffices, or add a more convincing experiment?
- In line 103, you mentioned your experiment contributes to "how lightweight feedback mechanisms empower smaller organisations to adopt DFL". Could you elaborate more specifically, along with your experiments? I don't find a very direct link there.
- In line 21, "transfers this knowledge between users to mitigate data sparsity": I don't think the data sparsity can be mitigated, but the decision and learning challenges due to data sparsity could be mitigated.

---

> ### Author Response · Authors · 2025-11-22
> **Response to Reviewer rr4b (pt 1/4)**
>
> Thank you for your extremely thorough and insightful review. Below, we answer each of your queries and indicate the specific sections where revisions have been made.
>
> **Literature**
> 1. ***The related works seem not to cover enough literature....***
>
> The mentioned works are indeed relevant to our topic and are included in the extended related works section in Appendix A lines 655-672. To our knowledge, there has been no effort from previous works (including those cited in the literature) to address noise and limited feedbacks/labels. Our work is the first to model Decision Focused Learning when only sparse binary feedback on decision outcomes is available. While some works do hint at generating pseudo-labels by considering non-optimal solutions as "negative examples"[1], even recent foundational papers on the topic [2] omit any mention of scenarios where a correlated dataset of unknown parameters is not available. More importantly, many practical applications open up when the assumption of the availability of the ground truth is removed. This is the main focus of our work.
>
> 2. ***In your setup, you have a latent assumption on the tolerance of decision makers that is a key structure for getting your theoretical results, thus you may also want to mention this compromising/tolerance effect from other literature, such as psychological/social sciences, to support and justify your setup.***
>
> We thank the reviewer for this insightful comment. We agree that the introduction of $\tau$ reflects an implicit assumption about how decision-makers provide feedback. This is in line with the concepts of "bounded rationality" and "satisficing" in preference modelling literature which posits that rational individuals will select a decision that is satisfactory rather than optimal. We have clarified this motivation in the revision, along with appropriate citations (See lines 152-155 in revised submission), explicitly noting that $\tau$ is used purely to interpret noisy feedback under a threshold. Our intention in using $\tau$ is not to model human behaviour, but rather to quantify the idea that positive feedback does not necessarily imply that an option is optimal, only that it is “good enough” relative to the user’s latent preferences. In our experiments conducted since the submission, we have found that the threshold $\tau$ seems to be quite robust. We observed that changing its value has little effect on the model's convergence. However, we do not yet have a theoretical guarantee regarding this robustness. Therefore, we have only included these experiments in the supplementary code material and refrain from making any claims about $\tau$ in the submitted text.
>
> **Model**
>
> 1. ***You mentioned "multi-agent" setting in your introduction, but the parameter $\theta^\*$ is assumed to be fixed for all agents...***
>
> As the reviewer recognised, we first outline the framework for a single agent before expanding it to multiple-agents. We have clarified this distinction at the beginning of the text for better understanding. (See lines 165-166 Section 3.1 in revised submission).
> Our approach is divided into two phases. In the first phase, we introduce the DFL framework, which relies on binary feedback on decision outcomes instead of traditionally used `ground truth' in DFL. The second phase addresses the issue of data scarcity. When a single agent does not generate enough data for effective learning, we can utilise the global data corpus from all agents to mitigate this scarcity.
>
> The motivation for introducing a multi-agent setting arises from the practical limitations encountered in real life when data from users is scarce (Section 1, lines 57-59, 72-76). Specifically, for a given problem, there may be a large number of agents or users, but each user has limited interactions. For example, the feedback dataset utilised in this study comprises data collected from over 12,000 users; however, over 91\% of these users have at most two interactions and fewer than 30 users have more than 10 interactions (Section 6.2 line 443-444, Appendix D.1.2 Figure 3a). This limited amount of data is insufficient for meaningful personalisation. However, as demonstrated in our results (Section 6.2 Figure 2), by inferring shared information from 'similar' users using collaborative training, we can still identify and model user preference groups effectively.
>
> [1] M. Mulamba, J. Mandi, M. Diligenti, M. Lombardi, V. Bucarey, and T. Guns, ‘Contrastive Losses and Solution Caching for Predict-and-Optimize’, in Proceedings of the Thirtieth International Joint Conference on Artificial Intelligence, Montreal, Canada: International Joint Conferences on Artificial Intelligence Organization, Aug. 2021, pp. 2833–2840. doi: 10.24963/ijcai.2021/390.
>
> [2] J. Mandi et al., ‘Decision-Focused Learning: Foundations, State of the Art, Benchmark and Future Opportunities’, J. Artif. Int. Res., vol. 80, Sept. 2024, doi: 10.1613/jair.1.15320.

---

> ### Author Response · Authors · 2025-11-22
> **Response to Reviewer rr4b (pt 2/4)**
>
> 2. ***What are the assumptions of the utility function $u(x, \theta)$ (e.g. shape of $u$ w.r.t. $x$ and $\theta$, continuity)?***
>
> The assumptions used to model $u(x, \theta)$ are outlined in Section 3.2 of the manuscript. In particular, $u(x, \theta)$ is assumed to be a sigmoidal function for the experiments conducted in this study. However, the framework is not restricted to only sigmoidal functions. For example, $u(x, \theta)$ could also be represented as simple logits, $\theta^\top x$. In this scenario, the binary cross-entropy (BCE) loss would be replaced with hinge loss, while the rest of the framework would remain unchanged.
>
> 3. ***In equation (2), I did not quite get it. Is the $\theta$ the ground-truth parameter?...***
>
> The notation for $\theta$ is standard practice in the Decision Focused Learning (DFL) literature [2]. Consider the optimisation problem $\arg\max_x u(x, \theta)$, where $u(x, \theta)$ is the objective function and $\theta$ is the parameter. If $\theta$ is known in advance, there is no need for a DFL solution, and the optimisation problem can be solved using the usual methods. However, if $\theta$ is unknown, the optimisation problem cannot be solved directly.
>
> The DFL framework aims to learn this $\theta$ using a correlated dataset, denoted as $\mathcal{D} = \\{(z_i, \theta_i)\\}_{i=1}^N$. This is achieved through a "prediction" model, $m(\omega, z)$, where the predicted parameter for the input-target pair $(z, \theta) \in \mathcal{D}$ is given by $\hat{\theta} = m(\omega, z)$. Unlike typical machine learning methods that focus on maximising the prediction accuracy of $\hat{\theta}$ in relation to $\theta$, the DFL framework aims to predict $\hat{\theta}$ such that the decision from the downstream optimisation problem, $x^\*(\hat{\theta}) = \arg\max_x u(x, \hat{\theta})$, is as close to the optimal solution as possible, where the optimal solution is defined as $x^\*(\theta) = \arg\max_x u(x, \theta)$.
>
> A measure of how close the decision based on the predicted parameter is to the optimal decision is called regret. Regret quantifies the loss in the objective value when using the predicted parameter compared to the ground truth parameter decision, calculated as $u(x^\*(\theta), \theta) - u(x^\*(\hat{\theta}), \theta)$ for some $\theta$ and the predicted $\hat{\theta}$.
>
> 4. ***Gradually following your description of the model, I sort of understand this is an online learning setup, but it is not stated clearly in the introduction...***
>
> In Section 3.1, lines 144-146, we begin with the assumption of a feedback dataset $\mathcal{D} = \\{(z_i, x_i^*, f_i)\\}_{i=1}^N$ rather than a continuous stream of feedback through an online learning mechanism. Our framework is adaptable to both online human-in-the-loop and offline predict-and-optimize scenarios. Importantly, the input to the model is not necessarily based on previous feedback.
>
> 5. ***Line 147-148: $f(x, \theta)$ is defined as a function w.r.t. both $x$ and $\theta$...***
>
> Thank you for pointing out the confusion with respect to the definition of 'ground-truth' and function $f$. We have revised section 3.1 to clarify this. The function $f(x, \theta)$ represents the feedback obtained on a decision $x$ from an oracle that has access to the true parameter $\theta$. However, as mentioned previously, we do not have access to $\theta$; it is a latent parameter. Thus, $f(x, \theta)$ is indeed a function of both $x$ and $\theta$. While $\theta$ represents the "ground truth", it should be understood in the context of traditional decision-making frameworks.
>
>  6. ***As you introduced below equation (2), the surrogate loss is to address minimizing the regret in non-trivial cases...***
>
> The reviewer is correct that the surrogate loss addresses both of these issues. Section 3.1 now addresses this more clearly.
> The problem of differentiating through an $\arg\max$ is, in fact, the primary challenge in a traditional DFL setting, which motivates the design of a differentiable surrogate loss, particularly when a correlated dataset (e.g., $\\{(z_i, \theta_i)\\}_{i=1}^N$) is available. Techniques such as Smart-Predict-and-Optimise aim to model a surrogate function for regret when this dataset is present.
>
> However, in our specific context, we encounter an additional problem of the lack of correlated data, which means we do not have access to $\theta$ itself. Our surrogate loss is intended to tackle both of these challenges. Furthermore, in our case, the input consists of $(z, f(x, \theta))$, and the discrete action set $\mathcal{X}\_z$ serves as the decision variable space for the optimisation problem $\arg\max_{x \in \mathcal{X}_z} u(x, \hat\theta)$ with respect to the input $z$ and the predicted parameter $\hat\theta = m(\omega, z)$.
>
> [2] J. Mandi et al., ‘Decision-Focused Learning: Foundations, State of the Art, Benchmark and Future Opportunities’, J. Artif. Int. Res., vol. 80, Sept. 2024, doi: 10.1613/jair.1.15320.

---

> ### Author Response · Authors · 2025-11-22
> **Response to Reviewer rr4b (pt 3/4)**
>
> 7. ***In line 174-179, I am not so convinced the choice of sigmoidal function is the right nonlinear mapping to be chosen...***
>
> The motivation for modelling the utility function as a sigmoid is a purely practical choice, as discussed in in Section 3.2. Particularly, as we do not have access to the actual ground-truth parameter $\theta$, we approximate the objective function by maximising the probability of receiving positive feedback ($f(x, \theta)$). Using a sigmoid function for this purpose is the simplest choice, as it simplifies the analysis and description of loss functions and the overall framework.
>
> In response to the typos regarding the noise and scale parameters $s$ and $b$ are indeed tunable hyperparameters. We select their values via Optuna during model selection/parameter tuning. We have made these corrections in the revision (line 186).
>
> 8. ***A follow-up question: Is the method sensitive to the design of the surrogate loss or hyperparameters?***
>
> Yes, the method is sensitive to the design of the surrogate loss. Developing an appropriate surrogate loss function to minimise regret is the main challenge in DFL. In most previous works, the choice of surrogate loss is either task-dependent or based on the specific nature of the optimisation problem [2]. However, there are some studies that aim to propose a loss function that is independent of these factors [3]. Despite this, there is still no consensus on a single, generalised loss function for DFL, and no existing surrogate loss designs are applicable to our task.
>
> The surrogate loss we propose in our work is simple in terms of its explainability and demonstrates strong performance based on empirical evidence. While there may be other possible surrogate loss definitions that could be better suited for this task, the core problem of using binary feedback for decision outcomes has not been explored in the context of DFL to the best of our knowledge. As a result, we lack established literature to conduct a comparative analysis of the design of our surrogate loss.
>
> 9. ***in line 197 \& 206, multiple $\mathcal{X}_t$ are not properly defined.***
>
> The typos have been corrected in the submitted revision. At line 144, the discrete decision space for the query $z$ is defined as $\mathcal{X}_z$, which is also used at line 206. Additionally, there is a typo at line 197; it should also read $\mathcal{X}\_z$.
>
> 10. ***The resolution of Figure 1 needs to be improved. Consider outputting PDF format instead***
>
> Thank you for pointing this out. The image resolutions will be improved in the final revision.
>
> 11. ***seems to Section 5.0.1 is a wrong index?***
>
> Section 5.0.1 is reformatted as a definition instead of "Section 5.0.1" in the revision. It was initially intended as a subsection related to the introductory paragraph of Section 5.
>
> 12. ***Can the approach be extended beyond binary feedback to other objective settings (e.g., ordinal, multiple target)?***
>
> To reiterate for the sake of brevity, our work assumes that instead of having a dataset structured as $\\{(z_i, \theta_i)\\}_{i=1}^N$, we have the dataset $\\{(z_i, x^\*_i, f(x^\*_i, \theta_i))\\}\_{i=1}^N$, where $f(x^\*_i, \theta_i)$ is a sparse binary feedback that serves as an abstraction of $\theta_i$ for $x_i$. As a result, if the target variable is represented by a value $\gamma \in \mathbb{R}$ instead of a binary outcome, our framework effectively simplifies to the traditional DFL framework. However, it is crucial to note that our current framework does not support multi-class labels. This limitation exists for two main reasons: first, the assumptions outlined in Section 4.1 are valid only for binary feedback; and second, the framework was initially designed as an approximation of DFL for cases where feedback is limited to simple like/dislike responses, rather than more informative numerical values.
>
> [2] J. Mandi et al., ‘Decision-Focused Learning: Foundations, State of the Art, Benchmark and Future Opportunities’, J. Artif. Int. Res., vol. 80, Sept. 2024, doi: 10.1613/jair.1.15320.
>
> [3] S. Shah, K. Wang, B. Wilder, A. Perrault, and M. Tambe, ‘Decision-Focused Learning without Decision-Making: Learning Locally Optimized Decision Losses’, Advances in Neural Information Processing Systems, vol. 35, pp. 1320–1332, Dec. 2022, Accessed: July 30, 2025. [Online]. Available: https://papers.nips.cc/paper_files/paper/2022/hash/0904c7edde20d7134a77fc7f9cd86ea2-Abstract-Conference.html

---

> ### Author Response · Authors · 2025-11-22
> **Response to Reviewer rr4b (pt 4/4)**
>
> **Experiments**
>
> 1. ***For your experiments in Section 6.1, I don't think you can produce meaningful conclusions with merely 5 data samples/users.***
>
> In our experiments, we limit each user to 5 data samples because our method is designed for extreme sparsity, where per-user data is minimal. Our goal is to assess whether the collaborative training framework can uncover a stable population-level structure from just a few interactions. This approach is relevant in real-world scenarios where gathering more than a few interactions per user is often impractical. While five samples may be too few for reliable individual models, our focus in on aggregating information from many users to create a statistically meaningful global dataset.
>
>  Collaborative training is particularly valuable in these sparse settings, allowing users with limited data to benefit from a shared structure. As more per-user data becomes available, reliance on collaborative learning would naturally decrease. Nevertheless, to ensure our conclusions are robust, we conduct extensive ablation studies, evaluate a large number of users, and report means and variances across multiple random seeds. Our results consistently show that, in this sparse-data context, our model significantly outperforms relevant baselines
>
> 2.  ***In line 103, you mentioned your experiment contributes to "how lightweight feedback mechanisms empower smaller organisations to adopt DFL"...***
>
> We aim to demonstrate that even a small number of samples per user can be sufficient for effective learning. Our goal is to show that organisations with a limited user base and insufficient interaction to generate large amounts of data can still acquire valuable insights to solve their optimisation problems. In essence, smaller organisations can utilise non-invasive information-gathering techniques through weak labels (thumbs up/thumbs down on decisions), enabling meaningful decision-focused learning even without access to large-scale data. The personalised trip planning algorithm suggested as a use case of such weakly supervised DFL is one such example.
>
> 3. ***In line 21, "transfers this knowledge between users to mitigate data sparsity": I don't think the data sparsity can be mitigated, but the decision and learning challenges due to data sparsity could be mitigated.***
>
> The reviewer correctly notes the distinction, which was the intention of the manuscript. This has been updated in the revision.
>
>
>
> We sincerely hope that we have been able to satisfy the reviewer's doubts to a large extent and have been able to demonstrate both novelty of the problem, efficacy of the solution and the impact of our work through the revised paper as well as the answers here.

---

### Meta-Review · Area_Chair_a5du · 2026-01-06

**Summary:**

Paper extends Decision-Focused Learning (DFL) to settings with sparse binary feedback. Authors design new learning framework for some instantiations of the problem from personalized trip planning (shortest path) and recommender systems space. Paper showcase increased performance of their approach which also uses fewer datapoint through experiments on large-scale proprietory datasets and one standard benchmark.

Problem and results are interesting. But presentation is lacks clarity and paper makes a lot of unjustified and often non-rigorous modeling choices and technical assumptions, e.g. eqns (3) and (5), line 209, and section 4.1. This was pointed out by multiple reviewers and AC also noted it in their independent reading of the paper. Reviewer also had questions computational cost and  hyper parameter sensitivity. A reviewer also notes that some of the experimental designs are not conclusive enough.

Additionally AC also notes that the technical analysis is very hand wavy. For example, probability of f is not properly defined and in Appendix C it is not clear why taking expectation over (z, \theta) integrates over all the randomness in f. It also not clear how Appendix B justifies the proposed loss function. From two given datapoints, it isn’t clear whether higher cost matter or not. There could be all sorts of reasons why someone prefer higher cost mode of transportation.

Finally, AC notes that authors potentially sidestepped original submission page limits by uses a incorrect citation format style. According to the style guide:
> Citations within the text should be based on the natbib package and include the authors’ last names and year (with the “et al.” construct for more than two authors).

**Reviewer Concerns:**

Authors addressed issues around computational cost and hyperparameter sensitivity. However, they failed to address the lack of clarity, rigour, and correctness in the technical exposition.

**Reviewer Scores:**

Assume that Reviewer 4C8D will keep their score at 6 and all other reviewers will keep or increase their scores to 4.

---

### Decision · Program_Chairs · 2026-01-26

Reject